



**Elucidating the pollution characteristics of nitrate, sulfate and ammonium in**
**PM2.5 in Chengdu, southwest China based on long-term observations**
Liuwei Kong[1], Miao Feng[2], Yafei Liu[1], Yingying Zhang[1], Chen Zhang[1], Chenlu Li[1], Yu
Qu[3], Junling An[3], Xingang Liu[1,*], Qinwen Tan[2,*], Nianliang Cheng[4], Yijun Deng[5],
Ruixiao Zhai[5], Zheng Wang[5]
[1]State Key Laboratory of Water Environment Simulation, School of Environment,
Beijing Normal University, Beijing 100875, China
[2]Chengdu Academy of Environmental Sciences, Chengdu 610072, China
[3]State Key Laboratory of Atmospheric Boundary Layer Physics and Atmospheric
Chemistry, Institute of Atmospheric Physics, Chinese Academy of Sciences, Beijing
100029, China
[4]Beijing Municipal Environmental Monitoring Center, Beijing 100048, China
[5]Yuncheng Municipal Ecological Environment Bureau, Yuncheng, 044000, China
* Corresponding author.
E-mail addresses: liuxingang@bnu.edu.cn (Xingang Liu) and 11923345@qq.com
(Qinwen Tan)
**Abstract**
Nitrate, sulfate and ammonium (NSA) are the main secondary inorganic aerosols of
PM2.5 and play an important role in the process of air pollution. However, few studies
have analysed the variation characteristics of NSA in PM2.5 and the effects of control
measures through long-term observations. In this study, a long-term observational
experiment was conducted from January 1, 2015 to December 31, 2017 in Chengdu,
southwest China. NSA pollution characteristics, chemical conversion generation,
emission reduction control sensitivity analysis and pollutant regional transport
characteristics were analysed. The concentrations of sulfate and ammonium in PM2.5
have been well reduced, but the effect of reducing nitrate was not obvious. Seasonal
and diurnal variations have obvious characteristics, winter still has a high NSA
concentration and emission intensity, and the concentration during the day was higher



than that at night. Although the workday concentration was slightly higher than the
weekend concentration, the difference was nonsignificant. The chemical conversion
characteristics of NSA formation were comprehensively analysed, and the aqueous
phase oxidation process plays an important role in the conversion of NOx, $SO_2$ and $NH_3$
to NSA. The ammonia-rich environment became increasingly obvious in the
atmosphere of Chengdu. Under these conditions, the sensitivity of NSA concentration
variation was analyses using the ISORROPIA-II thermodynamic model, and the results
show that by reducing NOx and $SO_2$ emissions, not only can reduce the nitrate and
sulfate in $PM_{2.5}$, but also help reduce the formation of ammonium nitrate and
ammonium sulfate to reduce ammonium. The results also show that while carrying out
NSA emission reduction, it is also possible to generate potential risks of changes in
aerosol pH. Combined with meteorological conditions and a potential source
contribution function (PSCF) analysis, local emissions and regional emissions of
pollutants are found to have important impacts on Chengdu's atmospheric environment.
This research result not only provides an assessment of the current atmospheric
emission reduction effect but also provides an important reference for determining
methods to further reduce the NSA concentration in atmospheric $PM_{2.5}$.
**Keywords:** Secondary inorganic aerosols; Long-term observations; Pollution
characteristics; Chemical conversions; Source analysis; Chengdu
**1 Introduction**
In recent years, with the rapid development of China's domestic economy and
acceleration of the urbanization process, energy consumption and pollutant emissions
have also increased, which increases the burden on the atmospheric environment, and
severe air pollution has become the focus of social concern (Liu et al., 2013b;An et al.,
2019;Fu et al., 2014;Zhao et al., 2017). When air pollution forms, $PM_{2.5}$ (aerodynamic
diameter less than 2.5 μm) mass concentrations can reach a higher pollution level,
which not only reduces atmospheric visibility but also carries a large number of toxic
species into the human lungs, increasing the risks of cardiovascular and cerebrovascular



diseases, as well as harming human health (Chang et al., 2018;Tie et al., 2009;Kong et
al., 2019;Zhao et al., 2018;Yang et al., 2015b). Nitrate, sulfate, ammonium, organic
matter and elemental carbon are the main components of $PM_{2.5}$, among which sulfate,
nitrate and ammonium (NSA) are the main secondary inorganic components in $PM_{2.5}$
(Ji et al., 2019;Zheng et al., 2016). NSA mainly originates from the secondary aerosols
produced by complex chemical reactions of NOx, $SO_2$and $NH_3$ from coal combustion,
vehicle exhaust emissions and agricultural sources (Liu et al., 2013a;Wang et al.,
2016;Tian et al., 2017).
Because China's current main energy consumption is still fossil fuels, which are widely
used in industry, vehicles and residentially, the emission reduction space of NSA is still
restricted by a large number of gaseous precursors of NSA (Zhao et al., 2018;Tong et
al., 2019). In addition, the chemical conversion of$NO_2$, $SO_2$ and $NH_3$ to form NSA is
still very complex. For example, photochemistry may affect the formation of NSA at
high solar radiation, and the homogeneous reaction may dominate the formation of NSA
in high relative humidity (Cheng et al., 2016;Sun et al., 2014;Wang et al., 2016;Ohta
and Okita, 1990). The formation of sulfate can increase the acidity of aerosols (Sun et
al., 2014). In contrast, the presence of $NH_3$ can play a role in neutralization and maintain
the acid-base balance of aerosols (Wang et al., 2016). If improper control measures are
taken in pollution reduction control, such as further ammonia emission reduction,
acidification of aerosols and environmental problems of acid rain are the likely result
(Liu et al., 2019c). In addition to the local emission of pollutants, regional transport is
also an important influencing factor. Determination of regional transport sources of
pollutants, taking regional joint prevention and control measures, and jointly reducing
the emissions of pollutants will enable better air control effects, particularly in the
Beijing-Tianjin-Hebei region of northern China (Chen et al., 2019a).
The characteristics of higher concentrations proportion of nitrate, sulfate and
ammonium in $PM_{2.5}$ were also found in other polluted areas in China, such as Beijing-
Tianjin-Hebei, the Yangtze River Delta, the Pearl River Delta, the Fenwei Plain,



Chengdu-Chongqing region (An et al., 2019;Li et al., 2017;Liu et al., 2019d). In
response to this situation, the Chinese government issued an Air Pollution Prevention
and Control Action Plan (2013-207) in 2013 to reduce pollutant emissions and improve
air quality (Ministry of Ecology and Environment of the People's Republic of China,
2013, last access: 12 February 2020). A large number of treatment measures have been
taken in coal combustion, motor vehicle emissions and phase out of outdated industrial
capacities, and by 2017, China's ambient air quality control measures have achieved
good results (Liu et al., 2019a;Chen et al., 2019b;Cheng et al., 2019;Li et al., 2019b).
In Beijing, $PM_{2.5}$, $NO_2$ and $SO_2$ decreased by 35.2%, 17.9% and 69.8%, respectively,
in 2017 compared with 2013 (Beijing Municipal Ecology and Environment Bureau,
2017, last access: 12 February 2020). In Chengdu, $PM_{2.5}$, $NO_2$ and $SO_2$ decreased by
42.3%, 15.9% and 64.5%, respectively, in 2017 compared with 2013 (Chengdu
Municipal Ecology and Environment Bureau, 2017, last access: 12 February 2020). To
continue to promote air quality improvement, the Chinese government launched the "
Three-Year Action Plan for Winning the Blue Sky Defense Battle" in 2018, which puts
forward stricter requirements on how to further promote the implementation of
emission reduction plans (the Sate Council, 2018, last access: 12 February 2020).
Through long-term observations, a comprehensive analysis of $PM_{2.5}$ chemical
composition and source characteristics is carried out to verify the current
implementation effects of emission reduction, and in-depth analyses of pollution
reduction control characteristics is of great significance for the next step in air pollution
control. However, observations with high time resolution are very rare, and the time
period of these atmospheric observations usually includes several pollution processes
or last for weeks or months; thus, it is difficult to analyse the long-term change
characteristics of air pollution through comprehensive observational means (Sun et al.,
2013;Tie et al., 2017;Guo et al., 2014). Especially in the Sichuan Basin of Southwest
China, there are few long-term observational experiments on NSA, which is the main
chemical component of $PM_{2.5}$.



The Sichuan Basin is among the most important areas of air pollution in China (Qiao
et al., 2019;Gui et al., 2019;Zhong et al., 2019). Although there are many studies in this
area, there are few long-term view studies of the hourly concentration data resolution
of PM$_{2.5}$ chemical components. In this study, through long-term observations (from 1
January 2015 to 31 December 2017), we analysed the pollution level and chemical
conversion characteristics of NSA in PM$_{2.5}$ in Chengdu and the concentration change
sensitivities of sulfate, nitrate and ammonium. Finally, combined with local emissions
and regional transmission characteristics, we analysed the regional transport
characteristics of Chengdu air pollution.
**2 Experiment and methods**
**2.1 Observation site**
Comprehensive observations were carried out at the Super Station of Atmospheric
Environmental Monitoring of Chengdu Academy of Environmental Protection
Sciences (30.65°N, 104.05°E). The site is located in the Wuhou District of Chengdu,
between First Ring Road and Second Ring Road (Fig. 1). This is a typical residential,
transportation and commercial mixed area that represent the characteristics of the urban
atmospheric environment. Chengdu is also a megacity in the Sichuan Basin of
Southwest China, as well as an important part of the Chengdu-Chongqing region, which
is among the regions with serious air pollution in China (Fig. 1).

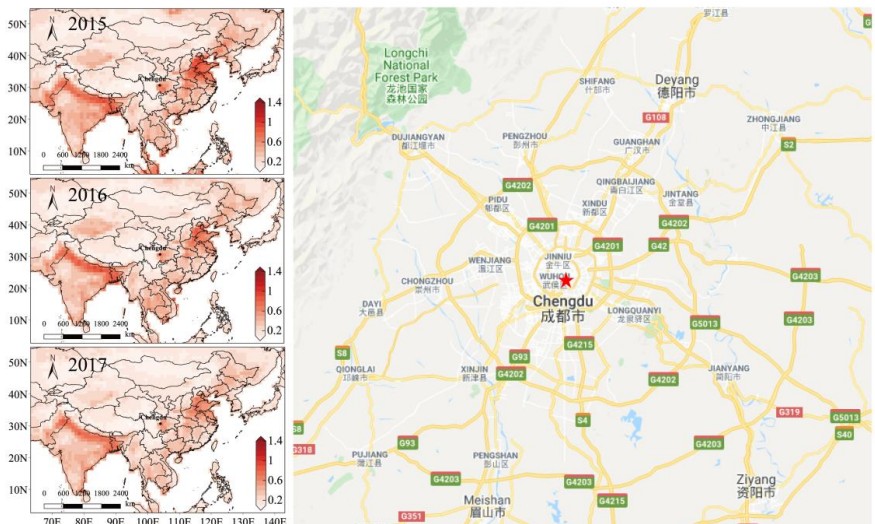


Fig. 1. Observation site in Chengdu. The image on the left shows the aerosol optical

depth (AOD, 550 nm) from 2015 to 2017 (National Aeronautics and Space

Administration, 2019, last access: 12 February 2020). The red star in the image on the

right shows the location of the observation site in Chengdu (© Google Maps 2020).

**2.2 Instruments**

During the research period, online experimental monitoring instruments were used to

obtain the observation data with an hourly resolution. The equipment list is shown in

Table 1.

Table 1. The experimental instruments used in this study

| Instrument Model | Parameters | Manufacturer/Country |
|---|---|---|
| URG-9000 | $NO_3^-/SO_4^{2-}/NH_4^+/Na^+/Mg^{2+}/Ca^{2+}/Cl^-/K^+$ | Thermo Fisher Scientific/USA |
| SHARP 5030 | $PM_{2.5}/PM_1$ | Thermo Fisher Scientific/USA |
| RT-4 | OC/EC | Sunset Laboratory/USA |
| Xact-625 | Fe/Mn | CES/USA |
| 450i/17i/42iY/48i/49i | $SO_2/H_2S/NOx/NO_2/NO/NH_3/ CO/O_3$ | Thermo Fisher Scientific/USA |
| WXT520 | Meteorological parameters | VAISALA/Germany |




**2.3 Chemical conversions and model methods**
To examine the conversion of gaseous pollutants to secondary aerosols, the sulfur
oxidation ratio (SOR) and nitrogen oxidation ratio (NOR) were used to reflect the
conversions of $SO_2$ and $NO_2$ to sulfate and nitrate, respectively (Sun et al., 2014;Yang
et al., 2015b). They can be calculated using Eq. (1) and Eq. (2):
$SOR=nSO_4^{2-}/(nSO_4^{2-}+nSO_2)$ (1)
$NOR=nNO_3^-/(nNO_3^-+nNO_2)$ (2)
where n is the molar concentration.
The ISORROPIA-II thermodynamic model was used to analyse the variation
characteristics of the interaction among aerosol chemical components (Fountoukis and
Nenes, 2007;Guo et al., 2017;Ding et al., 2019). Temperature (T), relative humidity
(RH) and the total concentrations (i.e., gas + aerosol) of $Na^+$, $SO_4^{2-}$, $NH_3$, $NO_3^-$, $Cl^-$,
$Ca^{2+}$, $K^+$ and $Mg^{2+}$ were input into the ISORROPIA-II thermodynamic model. In this
study, we use "forward problems" mode to run the model, assuming that the aerosols
were in a "metastable" state (salts do not precipitate under supersaturated conditions).
The simulated data and observed data were compared and analysed. Simultaneously,
the aerosol water content (AWC) and pH of aerosols were calculated. The sensitivity of
the interaction between aerosol chemical components (NSA) was analysed (Ding et al.,
2019;Fountoukis et al., 2009). The pH can be calculated using Eq. (3):
$pH = -log_{10}H_{aq}^+ \cong -log_{10}\frac{1000H_{air}^+}{AWC}$ (3)
where $H_{aq}^+$ (mol/L) is the concentration of hydronium ions in liquid water of
atmospheric particulate matter, which can be calculated by the $H_{air}^+$ and AWC (μg/m³)
outputs from the ISORROPIA-II thermodynamic model (Ding et al., 2019;Guo et al.,

166 2017).

Gas-particle phase partitioning can be used to describe the transformation
characteristics between semivolatile inorganic salts ($NO_3^-$ and $NH_4^+$) and corresponding
gases ($HNO_3$ and $NH_3$) (Guo et al., 2017), which can be calculated by Eq. (4) and Eq.

170 (5):


$\varepsilon(NO_3^-) = \frac{NO_3^-}{HNO_3 + NO_3^-}$        (4)
$\varepsilon(NH_4^+) = \frac{NH_4^+}{NH_3 + NH_4^+}$        (5)
where the units of $NO_3^-$, $NH_4^+$, $NH_3$ and $HNO_3$ were µg/m³, and the $HNO_3$ data are
from the ISORROPIA-II thermodynamic model output.
**2.4 CPF and PSCF methods**
To analyse the relationship between pollutants and wind direction (WD) and wind speed
(WS), the conditional probability function (CPF) was introduced the R Programming
Language. This function can be defined as CPF= $m_{\theta,j}/n_{\theta,j}$, where $m_{\theta,j}$ is the number of
samples in the WD interval $\theta$ and WS interval j with mixing ratios greater than some,
given a 'high' pollution concentration (percentile of pollutants), and $n_{\theta,j}$ is the total
number of samples in the same wind direction-speed interval (Uria-Tellaetxe and
Carslaw, 2014). Usually, a higher given 'high' pollution concentration (percentile) is
chosen, such as the 90th percentile, which will mask the lower percentile pollution
concentration source contributions. In this work, to obtain a more complete contribution
of pollution sources, a range of percentile values were selected for the CPF calculation,
e.g. 0-25, 25-50, 50-75 and 75-100.
The potential source contribution function (PSCF) is based on the analysis of pollution
sources, which is based on the airmass backward trajectory and can be used to judge
the long-distance regional transport of pollutants (Ji et al., 2019). In this study,
Meteoinfomap and TrajStat (Wang et al., 2009) were used, and the Hybrid Single-
Particle Lagrangian Integrated Trajectory (HYSPLIT) data input model were provided
by National Oceanic and Atmospheric Administration (National Oceanic and
Atmospheric Administration, 2019, last access: 12 February 2020); these data was used
for calculating the 24-hr backward trajectories at the observation site at a height of 500
m every 1 hr from 1 January 2015 to 31 December 2017 (UTC+8). The calculated
domain for PSCF was a range of 20-50° N, 75-115° E, and a grid cell with a resolution
of 0.5°×0.5° was divided. The PSCF can be defined as Eq. (6):



$$PSCF_{ij} = \frac{M_{ij}}{N_{ij}} W_{ij} \qquad (6)$$
$$W_{ij} = \begin{cases} 1.0 \ (N_{ij} \geq 3N_{ave}) \\ 0.7 \ (3N_{ave} > N_{ij} \geq 1.5N_{ave}) \\ 0.4 \ (1.5N_{ave} > N_{ij} \geq N_{ave}) \\ 0.2 \ (N_{ave} > N_{ij}) \end{cases} \qquad (7)$$
where $PSCF_{ij}$ is the value for the ijth grid cell, $M_{ij}$ is the total number of endpoints in
the ijth grid cell, with pollution concentrations at the observation site (30.65°N,
104.05°E) that are greater than a given threshold value (75 percentiles are selected for
gaseous pollutants). $N_{ij}$ is the number of backward trajectory endpoints that fall in the
ijth grid cell during the simulation period. To reduce the uncertainty in $N_{ij}$, an empirical
weight function $W_{ij}$ was introduced in Eq. (7), where $N_{ave}$ is the average of $N_{ij}$ during
the simulation period (Ji et al., 2019;Zhang et al., 2017;Wang et al., 2009).
**3 Results and discussion**
**3.1 Pollution characteristics of the interannual and entire observation periods**
The annual average mass concentration of NSA and its proportion in $PM_{2.5}$ are shown
in Table 2. The annual averages of $PM_{2.5}$ were 67.78, 71.88 and 59.68 $\mu g/m^3$,
corresponding to 2015, 2016 and 2017, respectively. However, the pollution of $PM_{2.5}$
in Chengdu was much higher than the annual secondary guideline value (35 $\mu g/m^3$,
Ambient air quality standards/GB3095-2012) and the World Health Organization
annual guideline value (10 $\mu g/m^3$). The same $PM_{2.5}$ pollution problem was also a
serious problem in Beijing and Nanjing (Ji et al., 2019;Zheng et al., 2019). The annual
average mass concentration of NSA also changed significantly, and the difference was
large. The Mann-Whitney U test showed that the variation in $NO_3^-$ was nonsignificant
($p > 0.05$), and $SO_4^{2-}$ and $NH_4^+$ had obvious significance from 2015 to 2017 ($p < 0.05$),
indicating that $NO_3^-$ had not decreased significantly, and there was an increase in 2017
compared to 2015. $SO_4^{2-}$ continues to decline, and $NH_4^+$ was also lower in 2017
compared to 2016. Notably, $SO_4^{2-}$ and $NH_4^+$ decreased significantly in 2017 compared
with 2015, but the variation in $NO_3^-$ was nonsignificant. Meanwhile, the annual
averages of $NO_3^-/SO_4^{2-}$ were 0.95, 1.02 and 1.45 for 2015, 2016 and 2017, respectively,





indicating that the contribution of vehicle emissions as a mobile source to $PM_{2.5}$ was
increased compared with that of coal combustion as a stagnant source (Li et al.,
2017;Wang et al., 2015). As shown in Table S1, from 2013 to 2017, the emissions of
$NO_2$ in Chengdu were obviously higher than those of $SO_2$, but $PM_{2.5}$, $NO_2$ and $SO_2$ all
showed downward trends, which benefited from the implementation of the Air
Pollution Prevention and Control Action Plan launched by the Chinese government,
and Chengdu also launched a more detailed pollution control plan in 2014 (the People's
government of Chengdu, 2014, last access: 12 February 2020).
Table 2. Comparison of annual mass averages ($\mu g/m^3$) and proportions (%) for NSA
from 2015 to 2017.

| | $NO_3^-$ | $SO_4^{2-}$ | $NH_4^+$ | $PM_{2.5}$ | $NO_3^-/PM_{2.5}$ | $SO_4^{2-}/PM_{2.5}$ | $NH_4^+/PM_{2.5}$ |
|---|---|---|---|---|---|---|---|
| 2015 | 9.13 | 10.37 | 6.14 | 67.78 | 0.129 | 0.165 | 0.088 |
| 2016 | 9.27 | 8.53 | 6.16 | 71.88 | 0.123 | 0.133 | 0.089 |
| 2017 | 9.17 | 6.88 | 5.01 | 59.68 | 0.141 | 0.132 | 0.079 |










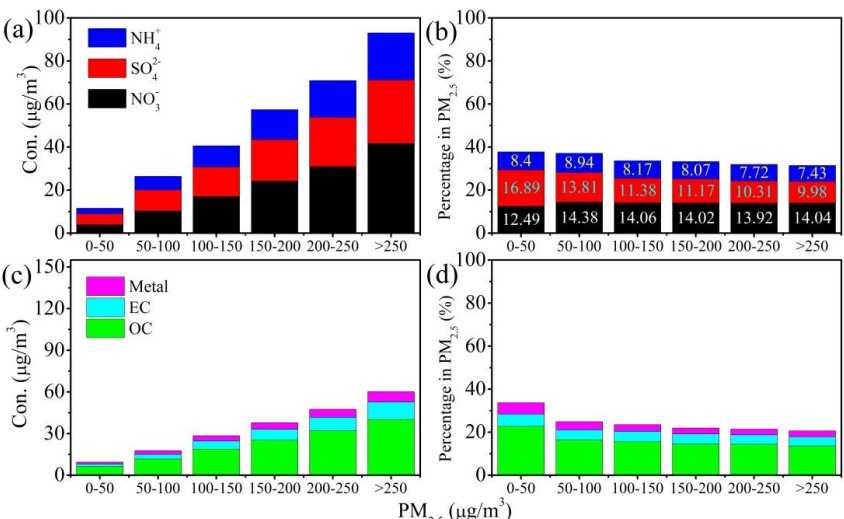

Fig. 2. Variation characteristics of the NSA and other chemical compositions with
different concentrations of $PM_{2.5}$. (a) NSA mass concentration. (b) Percentage of NSA
in $PM_{2.5}$. (c) Chemical compositions of organic carbon (OC), element carbon (EC), and
metal elements. (d) Percentage of other chemical compositions in $PM_{2.5}$.
The chemical composition of $PM_{2.5}$ from 2015 to 2017 varies with its concentration, as
shown in Fig. 2. With the accumulation of $PM_{2.5}$ in the atmosphere, the concentration
of NSA has also increased significantly, but their proportion in $PM_{2.5}$ has a downward
trend (Fig. 2a and b). When the $PM_{2.5}$ was less than 50 $\mu g/m^3$ and greater than 250
$\mu g/m^3$, the mass concentrations of NSA were 11.57 and 90.06 $\mu g/m^3$, respectively, and
the proportions were 37.78 and 31.45% respectively. Comparing Fig. 2b and d, it was
found that NSA was always the main contributor in the entire process of $PM_{2.5}$
accumulation, which was significantly higher than the proportions of OC and EC (Ji et
al., 2019;Li et al., 2019c). In the accumulation process of $PM_{2.5}$ concentrations greater
than 50 $\mu g/m^3$, nitrate accounts for a high proportion in NSA and was stable at
approximately 14%, and the proportion of sulfate and ammonium continues to decrease
(Li et al., 2019c;Wang et al., 2016). When the $PM_{2.5}$ concentration was less than 50
$\mu g/m^3$, the concentration of $SO_4^{2-}$ was higher than that of $NO_3^-$, and the concentration



of NH$_4^+$ was lower than the NH$_4^+$ concentration of PM$_{2.5}$ at 50 to 100 μg/m$^3$, possibly
due to sulfate concentration was higher than nitrate, forming more chemically stable
ammonium sulfate (Guo et al., 2017). In addition, when PM$_{2.5}$ was less than 50 μg/m$^3$,
low RH and strong solar radiation were also important ways to generate sulfate (Yao et
al., 2018).
**3.2 Monthly and seasonal variations**
The monthly variation characteristics of NSA from 2015 to 2017 are shown in Fig. 3.
These variations have similar trends due to meteorological factors (Fig. S1); from April
to August, they have higher temperature and WS, lower RH and atmospheric pressure,
which is not only conducive to the dilution and diffusion of pollutants but also reduces
the chemical conversions of pollutants by aqueous phase and the concentrate ability of
gaseous pollutants to concentrate particles (Wang et al., 2016;Ji et al., 2019). Overall,
the concentrations are higher in January and December and lower in July and August.
The highest monthly average of NO$_3^-$ reached 19.98 μg/m$^3$ in January 2017, and the
highest monthly average of SO$_4^{2-}$ and NH$_4^+$ were 22.08 μg/m$^3$ and 12.66 μg/m$^3$ in
January 2015, respectively. The lowest concentrations of NSA appeared in August 2017,
which were 1.96, 3.07 and 1.62 μg/m$^3$. The gaseous precursors of NSA also have
obvious monthly variations, and the NOx and SO$_2$ trends were similar to those of nitrate
and sulfate (Fig. 3 and S2). NH$_3$ emissions were significantly different, with increases
in warmer months (April-July) and colder months (September-December). On the one
hand, NH$_3$ volatilization was promoted by relatively high temperatures; on the other
hand, the use of agricultural fertilizers and livestock farming were also important
sources of NH$_3$ in China. Second, from urban areas, fossil fuel combustion and motor
vehicle emissions also contribute significantly (Liu et al., 2013a;Pan et al., 2016).
Notably, NH$_3$ increased significantly from April to December 2017 compared with
2015 and 2016, especially during low-temperature months (Fig. S2c). This also shows
that the implementation of air pollution reduction measures should increase the
emission reduction intensity in terms of NOx and NH$_3$ emissions, especially the





implementation of autumn and winter air pollution prevention and control action. The
seasonal variation in NSA was shown in Fig. S3, and the concentration in winter was
much higher than that in summer. Nitrate only declined in spring and summer from
2015 to 2017, with an increase in autumn and winter (Fig. S3a). Seasonal variations in
ammonium were similar to those of nitrates, with higher concentrations in winter and
the lowest in summer. This may be because higher temperatures and WSs not only can
promote the decomposition of ammonium nitrate in summer but also promote the
dilution and diffusion of pollutant concentrations (Guo et al., 2017;An et al., 2019).
Sulfate has a significant downward trend in all seasons from 2015 to 2017, especially
in winter. This downward trend was due to implementation of the Air Pollution
Prevention and Control Action Plan, especially the measures of "electricity instead of
coal" and "natural gas instead of coal" (refers to increased use of electricity and natural
gas in the residential sector to reduce coal combustion). The variation amplitude of NSA
and gaseous pollutants in cold months was significantly higher than that in warm
months (Figs. 3, S2 and S3). This higher variation amplitude may be due to the
differences in pollutant accumulation and scavenging processes. This finding also
indicates that the instability of local pollutant emissions and regional transport during
cold months was affected by meteorological conditions (Li et al., 2017;Ji et al., 2018).
The large variation amplitude of pollutants in different months, similar to the changes
in the Beijing-Tianjin-Hebei region of northern China and Chengdu, are due to the
accumulation and removal of pollution by meteorological conditions and pollutants
emissions (Ji et al., 2019;Qin et al., 2019;Zhang et al., 2019).



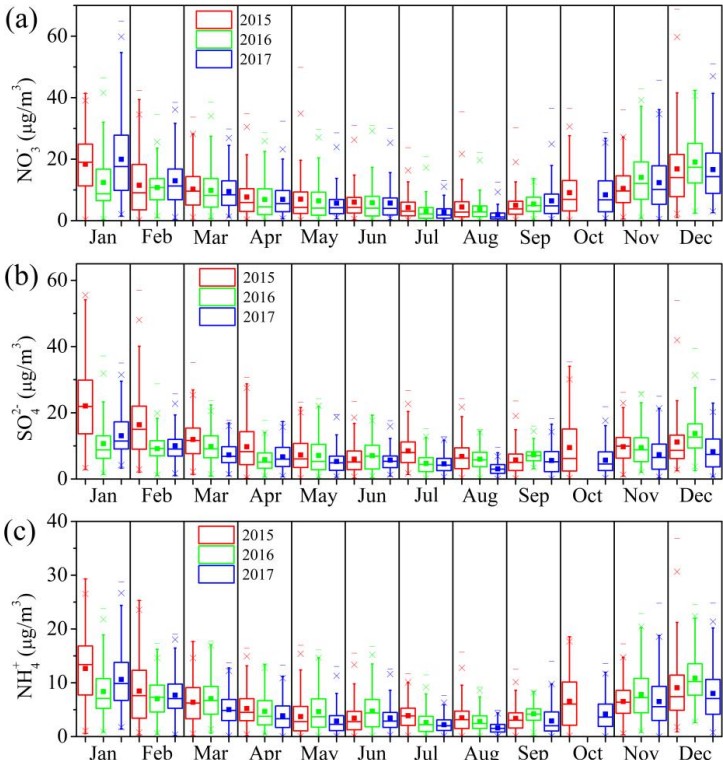


**Fig. 3. Monthly variations in NO₃⁻, SO₄²⁻ and NH₄⁺ mass concentrations from 2015**


**Fig. 3. Monthly variations in $NO_3^-$, $SO_4^{2-}$ and $NH_4^+$ mass concentrations from 2015**

**to 2017.3.3 Diurnal and weekly variations**
The diurnal variation in NSA is shown in Fig. 4; a similar trend was shown for daily
changes of nitrate, sulfate and ammonium, which was higher in the daytime than in the
evening. From 2015 to 2017, the diurnal variation trend of nitrate was similar, sulfate
was obviously reduced, and the ammonium was only significantly reduced in 2017. The
decrease in $NH_4^+$ may be closely related to the decrease in $SO_4^{2-}$ (Fig. 4a). Some studies
have shown that $NH_4^+$ in aerosols will first combine with $SO_4^{2-}$ ions and $HSO_4^-$ to form
$(NH_4)_2SO_4$ or $NH_4HSO_4$ and then combine with $NO_3^-$. The significant drop in $NH_4^+$ in
2017 may be due to a decrease in $SO_4^{2-}$. Similarly, the $NH_4^+$ did not show a significant
decrease in 2016, probably due to the increase in $NO_3^-$ in 2016 (Table 2), combined
with a portion of the $NH_4^+$ (Meier et al., 2009;Sun et al., 2014;Ding et al., 2019). This



finding also indicates that the concentration of $NH_4^+$ in particulate matter in Chengdu
may be affected by the concentration of $SO_4^{2-}$. From 2015 to 2017, the concentration of
NSA was higher in the daytime than in the evening (Fig. 4a), and similar results were
found in different seasons (Fig. 4b), which may be due to the combination of pollutant
emissions and meteorological conditions. As shown in Fig. S4, from 9:00 to 11:00 a.m.,
the concentrations of $SO_2$, NOx, $NH_3$, CO and other gases increased significantly,
indicating that the primary emission of pollutants was relatively strong. At this time,
higher RH (Fig S5) also provides favourable conditions for the formation of secondary
aerosols and promotes the accumulation of NSA (Cheng et al., 2016;Wang et al.,
2016;Sun et al., 2014). In addition, before 10 o'clock, relatively low WS will enable
easy pollutant concentration accumulation. In contrast, the higher WS in the afternoon
may be the main factor for the decrease in pollutant concentration (Fig. 4, S4 and S5).
Photochemical reactions may also be one of the factors in the formation of NSA, and
the concentration of $O_3$ peaks at approximately 15:00, which may be affected by the
free radicals generated by photochemistry. At approximately 19:00, the ratio of
$NO_2/NO$ reached its highest value, and the concentration of $NO_2$ also increased
significantly (Song et al., 2018;Zhu et al., 2019). At night, with the increase in RH (Fig.
S5), dissolved ozone, free radicals, hydrogen peroxide and $NO_2$ can catalyse $SO_2$ to
form secondary aerosols through an aqueous phase reaction (Zhang et al., 2015;An et
al., 2019). The seasonal diurnal variation in NSA was shown in Fig. 4b. The
concentration of NSA in winter was obviously higher than that in summer, and the
diurnal variation range was larger. The concentration in spring and autumn was closer,
but the diurnal variation in spring was larger than that in autumn. The larger diurnal
variation range not only indicates serious pollution but also indicates the importance of
other factors affecting air quality, such as meteorological conditions, secondary aerosol
conversion conditions, and so on (Ji et al., 2019;Yang et al., 2015a). The peak value of
the NSA seasonal diurnal variation also varies in different seasons. The peak value
appears at approximately 13:00 in winter, approximately 10:00 in spring and summer,
and approximately 12:00 in autumn, possibly due to the influence of meteorological
conditions.

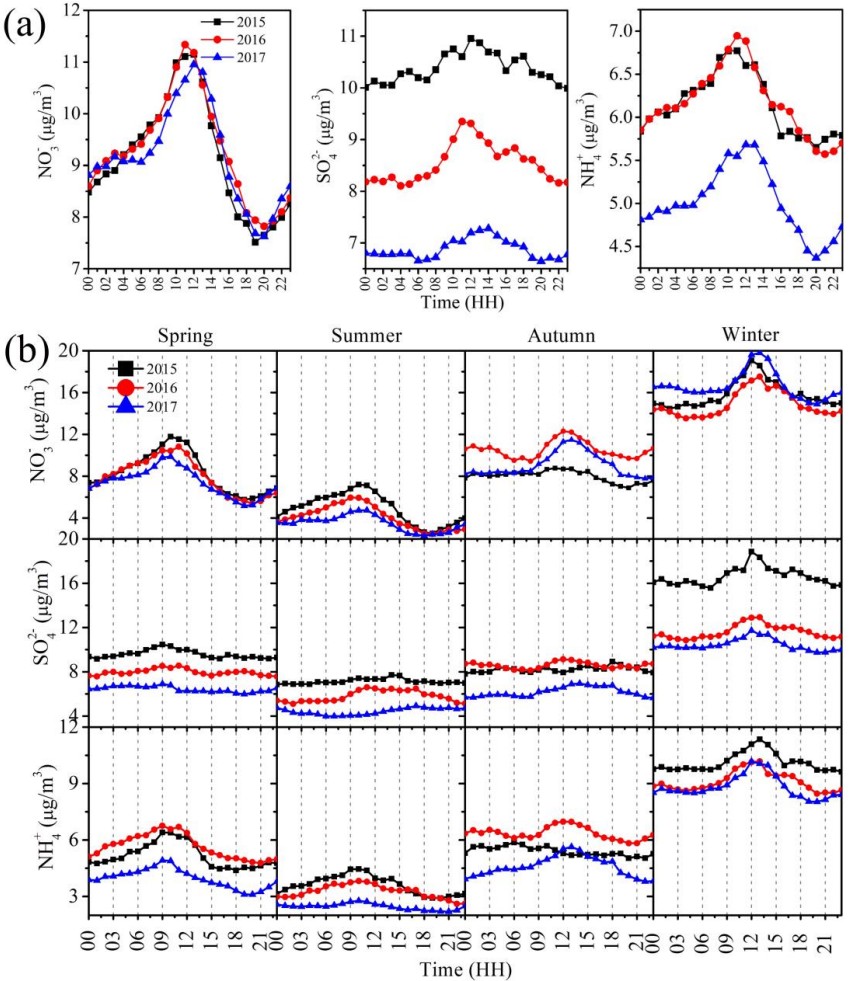


Fig. 4. Diurnal variations in NSA from 2015 to 2017. (a) Annual average. (b) Seasonal
average.
The weekly variation in NSA is shown in Figs. S6-8. During the overall observational
period, workdays (Monday to Friday) showed higher variations than the weekend
(Saturday and Sunday), with the highest variation being on Tuesday and the lowest
being on Sunday. Despite the difference in mean values between Tuesday and Sunday,
nonparametric tests show that the difference in mean values was nonsignificant (Mann-





Whitney U test, P > 0.05). As shown in Figs. S6-7, the average trends of $NO_3^-$ and $NH_4^+$
were consistent from Monday to Sunday. The correlation coefficient was 0.94 (P <0.01)
from 2015 to 2017, which indicates that they have a common source and that the vehicle
emissions also have an important contribution to $NH_4^+$ (Pan et al., 2016). The average
values of $NO_3^-$, $SO_4^{2-}$ and $NH_4^+$ from 2015 to 2017 were 9.21, 8.64 and 5.64 ug/m$^3$ on
workdays and 8.56, 8.33 and 5.29 ug/m$^3$ on weekends, respectively. Similarly, the
Mann-Whitney U test showed no significant difference. Population standard deviation
comparisons of $NO_3^-$, $SO_4^{2-}$ and $NH_4^+$ showed that workdays were higher than
weekends, with 7.96, 6.04 and 4.35 on weekdays, 6.76, 5.69 and 3.88 on weekends,
respectively (Fig. S8). Compared with the diurnal variations on weekdays and
weekends, the variations in nitrate and ammonium were more obvious than those of
sulfate (Fig. S9). In Beijing, a vehicle restriction scheme based on motor vehicle license
plates was implemented, that is, there are no restrictions on weekends, and the
contribution of vehicle emissions pollution on weekends was lower than that on
workdays (Ji et al., 2019). Similarly, Chengdu also implemented restriction measures
according to the license plate of vehicles on weekdays, but the average concentration
of pollutants on weekdays was slightly higher than that on weekends (Mann-Whitney
U test, P > 0.05). This finding shows that while implementing a policy of motor vehicle
restriction, improving the emission standards of motor vehicles and the quality of
gasoline and diesel oil was an important measure.
**3.4 Chemical characteristics of NSA**
**3.4.1 Relationship between NSA and carbonaceous components**
Precursor gases of NSA, such as $SO_2$, NOx and $NH_3$, are usually derived from coal
combustion, vehicle exhaust and agricultural sources, and they are accompanied by
emissions of carbonaceous aerosols (Wang et al., 2016). Fig. 5a, b and c show the
relationship between NSA concentration and CO, OC and EC, demonstrating a good
Pearson's correlation (p<0.01), which indicates that the emissions of carbon aerosols
were accompanied by the emissions of NSA precursor gases; these gases form NSA



through complex chemical reactions, such as photochemical, aqueous chemical
conversions and heterogeneous reactions (An et al., 2019;Li et al., 2019a;Zhu et al.,
2019). CO and EC usually originate from combustion sources, while OC originates
from primary emissions and secondary conversion (Tie et al., 2017;Tao et al.,
2017;Kong et al., 2019;Wu et al., 2016). The OC/EC value can be used to determine
the sources of carbon aerosols, such as vehicle exhaust, coal combustion and biomass
burning (Zhang et al., 2010). As shown in Fig. 5d, when the concentration of nitrate
and ammonium reached a peak, the OC/EC value was between 2-3, which was lower
than the OC/EC value when the sulfate was at the peak (3-4). Previous studies have
also shown that the OC/EC value of vehicle emissions was lower than that of coal
combustion (Cao et al., 2005;Kopp and Mauzerall, 2010). Nitrate and ammonium also
have similar trends, and their Pearson's correlation was 0.92 (p<0.01), which was
higher than that of ammonium and sulfate (0.88). The correlation coefficients of $NH_3$
with NOx and $SO_2$ were 0.42 and 0.23, respectively, suggesting that vehicle emissions
may also be a major source of ammonia (Pan et al., 2016;Liu et al., 2013a).

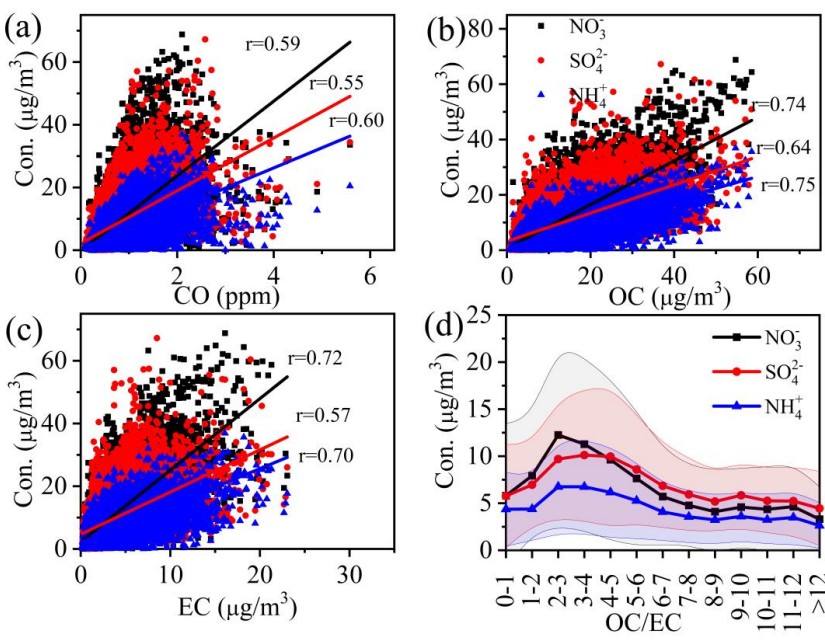






Fig. 5. Relationships between NSA and CO and OC and EC. (a) NSA and CO. (b)
NSA and OC. (c) NSA and EC. (d) NSA and OC/EC.
**3.4.2 Chemical conversion characteristics of NSA**
Figure 6 shows the abilities of $SO_2$ and $NO_2$ to chemically convert to sulfates and
nitrates and the variation trend of ozone concentration and metal elements at different
$PM_{2.5}$ concentrations. With the increase in $PM_{2.5}$ concentration, SOR and NOR
gradually increased, indicating that the formation ability of sulfate and nitrate increased
during the formation of air pollution. As the concentration of $PM_{2.5}$ increases, the
extinction properties of aerosols increase, the photochemical reaction conditions
weaken, and the $O_3$ concentration shows a decreasing trend, as shown in Fig. 6a. With
the accumulation of $PM_{2.5}$ concentration, metal elements (Fe and Mn) also showed an
increasing trend and were similar to SOR and NOR. Previous studies have shown that
mineral dust elements such as Fe and Mn can play a catalytic role in the formation of
atmospheric sulfate (Martin and Good, 1991;He et al., 2014). The Pearson's correlation
statistics of SOR and NOR with Fe and Mn under different $PM_{2.5}$ concentration
conditions are shown in Table S2; it is only under high $PM_{2.5}$ concentration conditions
(>200 µg/m$^3$) that SOR and NOR have a positive correlation. This result is similar to
those of previous studies in Beijing and Xi'an, where Fe and Mn play a limited catalytic
role in sulfate formation (Cheng et al., 2016;Wang et al., 2016). Some studies suggest
that when SOR is greater than 0.1, there may be a photochemical reaction pathway
leading to the conversion of $SO_2$ to sulfate (Ohta and Okita, 1990). Fig. 6a shows that
in addition to the photochemistry contributing to $SO_2$ oxidation, there may be a more
important pathway leading to the conversion of $SO_2$ to sulfate.

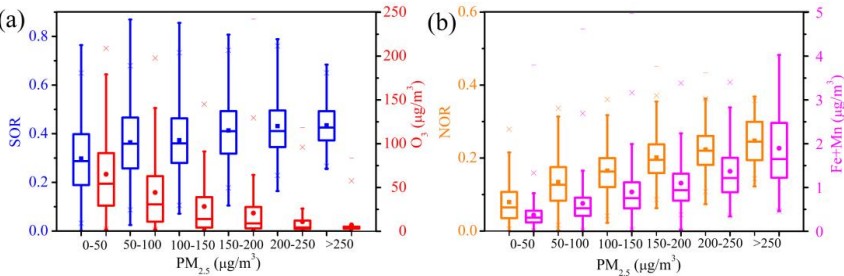


Fig. 6. Analysis of atmospheric chemical conversion ability at different $PM_{2.5}$
concentrations. (a) SOR and $O_3$. (b) NOR and concentration of metal elements (Fe and
Mn).
Figure 7 shows the variation characteristics of NSA chemical conversions and
meteorological conditions with increasing RH. SOR and NOR increased with
increasing RH, suggesting that $SO_2$ and $NO_2$ were more likely to produce sulfate and
nitrate under higher RH conditions. Previous studies have shown that in the presence
of $NH_3$, $NO_2$ can promote the chemical conversion of $SO_2$ to sulfate in the aqueous
phase (Wang et al., 2016). In aerosol water, alkaline aerosol components can promote
the dissolution of $SO_2$ and formation of sulfate under the oxidation of $NO_2$ (Cheng et
al., 2016). Especially when the atmosphere was polluted, the formation of sulfate by
$SO_2$ through the aqueous phase environment can contribute most of the sulfate (Sun et
al., 2013). According to the ISORROPIA-II thermodynamic model simulation, AWC
and pH also increase with RH (Fig. 7c and d). The increase in AWC can dilute the
concentrations of sulfate and hydrogen ions and promote an equilibrium shift in the $SO_2$
to sulfate during the aqueous phase. The increase in RH and gradual decrease in T can
also affect the gas-particle phase partitioning of $HNO_3$-$NO_3^-$ (Fig. 7g) and $NH_3$-$NH_4^+$
(Fig. 7h), prompting more nitrate and ammonium to condense in aerosol liquid water.
By comparing the NOR with the meteorological conditions and gas-particle distribution
when RH is greater than 80% (Fig. 7b, e, f and g), the increase in T and the decrease in
atmospheric pressure were not conducive to the conversion and presence of nitrate in
the aqueous phase (Guo et al., 2017;Ding et al., 2019). Therefore, Figs. 6 and 7 also



illustrate that the aqueous phase oxidation environment may contribute to the
generation of a larger portion of NSA.

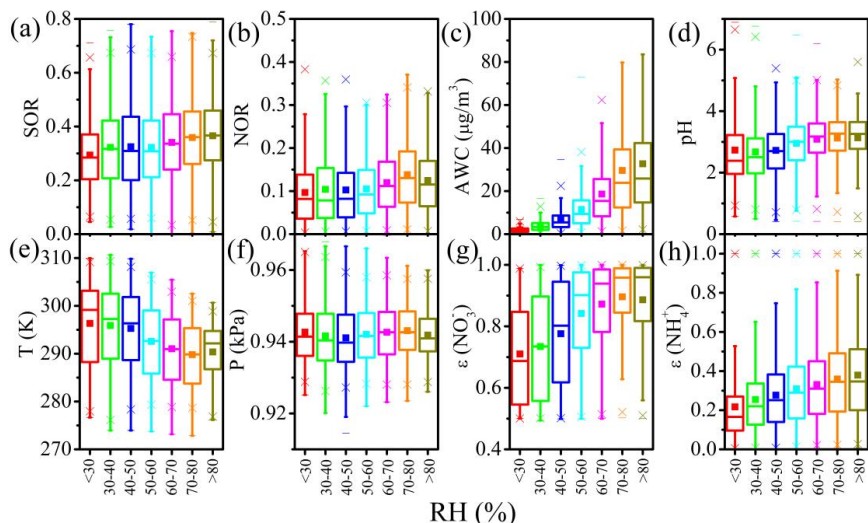

Fig. 7. Effects of RH on the chemical conversion of NSA. (a) SOR. (b) NOR. (c) AWC.
(d) pH of $PM_{2.5}$. (e) Temperature (T). (f) Atmospheric pressure (P). (g) $NO_3^-$ gas-particle
phase partitioning. (h) $SO_4^{2-}$ gas-particle phase partitioning.
**3.4.3 Sensitivity analysis**
The molar ratio analysis of NSA shown in Fig. 8 was used to analyse the chemical
relationships among NSA. $(NH_4)_2SO_4$ and $NH_4NO_3$ are mainly composed of $NH_4^+$,
$SO_4^{2-}$ and $NO_3^-$ in particulate matter (Malm and Hand, 2007;Meier et al., 2009).
Because $(NH_4)_2SO_4$ has better stability than $NH_4NO_3$, $NH_4^+$ will first combine with
$SO_4^{2-}$ and then with $NO_3^-$ (Liu et al., 2012). The annual average molar ratio of $NH_4^+$ to
$2*SO_4^{2-}$ was more than 1, which indicates that $SO_4^{2-}$ can be completely neutralized by
$NH_4^+$ (Fig. 8a). The molar ratios of residual $NH_4^+$ ($NH_4^+ - 2*SO_4^{2-}$) to $NO_3^-$ were 0.93,
0.99 and 1.05 in 2015, 2016 and 2017, respectively. As shown in Fig. 8a and b, the
gradual increase in the ratio (slope k) from 2015 to 2017 indicates an increase in
ammonia emissions from aerosols, especially in 2017, with a ratio of 1.05, indicating
the presence of other forms of ammonium salts, such as $NH_4Cl$ and $(NH_4)_2C_2O_4$ (Sun



et al., 2006). Seasonal variations in $NH_4^+$, $SO_4^{2-}$ and $NO_3^-$ are shown in Fig. 8c and d.
The higher molar ratio in autumn indicates that the intensity of ammonia emission in
autumn was higher than that in other seasons. This finding also shows that the problem
of atmospheric ammonia-rich environments in Chengdu in 2017 and autumn was more
prominent.

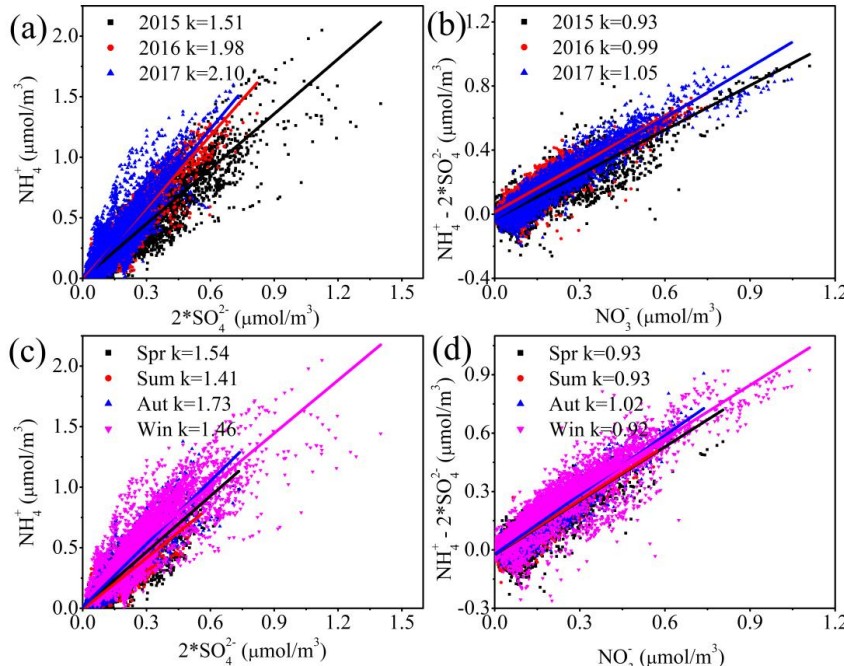


Fig. 8. Molar ratio analysis of NSA. (a) Interannual variation in the molar ratio of $SO_4^{2-}$
and $NH_4^+$. (b) Interannual variation in the molar ratio of $NO_3^-$ and $NH_4^+$. (c) Seasonal
variation in the molar ratio of $SO_4^{2-}$ and $NH_4^+$. (d) Seasonal variation in the molar ratio
of $NO_3^-$ and $NH_4^+$.
Table 3 shows the sensitivity analysis of the concentration variations in $SO_4^{2-}$, $NO_3^-$ and
$NH_4^+$. ISORROPIA-II thermodynamic model sensitivity analysis is described in detail
in the Supplementary Materials. The coefficient of variance represents the response of
the species to variations in other components. The coefficients of variance for $NH_4^+$ and
$NO_3^-$ produced by $SO_4^{2-}$ changes were 52.62 and 5.38, respectively. Similarly, the





coefficients of variance for $NH_4^+$ and $SO_4^{2-}$ produced by nitrate changes are 49.27 and
0.002, respectively. The large coefficient of variance for $NH_4^+$ indicates that the changes
in $SO_4^{2-}$ and $NO_3^-$ can affect the presence of $NH_4^+$, which also indicates that $(NH_4)_2SO_4$
and $NH_4NO_3$ were the main states of $NH_4^+$ (Liu et al., 2012). The coefficients of
variance for $SO_4^{2-}$ and $NO_3^-$ produced by $NH_3$ changes are 2.48 and 31.30, respectively,
which indicates that $NH_4^+$ was excessive to sulfate and that $NH_4^+$ first combines with
sulfate to form stable $(NH_4)_2SO_4$, and the remaining $NH_4^+$ and $NO_3^-$ will combine to
form $NH_4NO_3$. From 2015 to 2017, the coefficient of variance for $NH_4^+$ and $NO_3^-$
caused by the changes in $SO_4^{2-}$ gradually decreased, which may be attributed to the
decrease in $SO_4^{2-}$ concentration in $PM_{2.5}$ (Table 2). The coefficients of variance for $NO_3^-$
caused by the changes in $NH_3$ in 2015 and 2016 were 32.83 and 38.24, respectively. At
this time, $NO_3^-$ can completely neutralize $NH_4^+$. In 2017, the coefficient of variance
was 21.88, and the ammonia was a surplus (Fig. 8b); thus, the coefficient of variance
may be affected by the thermal instability of $NH_4NO_3$ during this time (Ansari and
Pandis, 2000;An et al., 2019). In terms of seasonal variation, the changes in sulfate and
nitrate can cause larger coefficients of variance for $NH_4^+$. When $NH_3$ changes, the
coefficient of variance for $NO_3^-$ was greater than that of $SO_4^{2-}$. In summer, the
coefficient of variance for $NO_3^-$ and $NH_4^+$ caused by the changes in $SO_4^{2-}$ were
obviously higher than those in other seasons. On the one hand, this may be due to the
relatively low concentrations of $NO_3^-$ and $NH_4^+$ in $PM_{2.5}$ due to lower gas-particle phase
partitioning in summer (Fig. S10a and b). On the other hand, the stronger
photochemical reaction may also lead to a greater change in the concentrations of $NO_3^-$
and $NH_4^+$ (Ohta and Okita, 1990). The coefficients of variance for $NO_3^-$ and $SO_4^{2-}$ in
winter were 6.13 and 0.005, respectively, which are higher than those in spring and
autumn, most likely due to higher NOR and SOR (Fig. S10c and d). Previous studies
have shown that the conversion of $SO_2$ to sulfate in the aqueous phase not only increases
the conversion of sulfate but also enhances the formation of nitrate particles in the
aqueous phase (Wang et al., 2016). Therefore, sulfate emission reduction may play a





key role in the process of controlling emission reduction in NSA pollution, as it not
only reduces the presence of $NH_4^+$ (($NH_4$)$_2SO_4$) in particulate matter but also affects
the formation of $NH_4NO_3$ by influencing the formation of nitrate. $NO_2$ and $NH_3$ can
also promote the conversion of $SO_2$ to sulfate through an aqueous phase environment
(Wang et al., 2016). Therefore, in the current ammonia-rich environment, priority
control of $SO_2$ and $NO_2$ emissions is an important way to reduce NSA in particulate
matter. Through the implementation of the Air Pollution Prevention and Control Action
Plan, the reduction in sulfate emissions has achieved good results. Therefore, while
continuing to promote "electricity instead of coal" and "natural gas instead of coal" to
reduce coal combustion pollution, more stringent control measures should be added for
nitrate and ammonia emissions. To further improve air quality, the Chinese government
launched a "Three-Year Action Plan for Winning the Blue Sky Defense Battle" in 2018
(the Sate Council, 2018) and proposed emission reduction targets for $SO_2$ and NOx
emissions, which will be 15% lower in 2020 than in 2015. By using the ISORROPIA-
II thermodynamic model to simulate $SO_4^{2-}$, $NO_3^-$ and $NH_3$ emission reduction control
effects of 5%, 10%, 15% and 20% respectively, the results were shown in Table S3.
The results show that a better effect can be achieved by controlling the $SO_4^{2-}$ and $NO_3^-$
emissions reduction, especially the effects of synergistic emissions reduction.
In addition, NSA can increase the hygroscopicity properties of aerosols, and more AWC
can increase the pH by diluting the hydrogen ion concentration (Kong et al., 2020;Ding
et al., 2019). Higher sulfates, nitrates and AWC correspond to a lower pH, indicating
that higher sulfates and nitrates have a greater effect on increasing aerosol acidity than
AWC dilution (Fig. S11a and b). Previous studies have also shown that sulfate
formation reduces aerosol pH (Sun et al., 2014). The same increase in ammonia
emissions can increase the aerosol pH (Fig. S11c). Table S3 also shows the impacts of
$SO_2^-$, $NO_3^-$ and $NH_3$ emissions reduction control on pH, such as sulfate and nitrate
emissions reduction increasing pH and $NH_3$ emissions reduction reducing pH, and
synergistic emission reduction has the least impact on pH changes, so controlling the



emissions reduction ratio in the air pollutant emission reduction scheme to reduce the
impacts of aerosol pH is worth further study. Acid rain is mostly concentrated in
southern China, and there are also important acid rain problems in the Sichuan Basin
(Fig. S12). Therefore, while controlling NSA emissions, especially controlling
ammonia emissions, the potential environmental problems of acid rain are worth
comprehensive assessment and analysis (Liu et al., 2019c).
Table 3. Sensitivity analysis of NSA concentration variations during the different
observation periods.

| Periods | Variables | Coefficient of variance | | |
|---|---|---|---|---|
| | | $NO_3^-$ | $SO_4^{2-}$ | $NH_4^+$ |
| 2015-2017 | $NO_3^-$ | | 0.002 | 49.27 |
| | $SO_4^{2-}$ | 5.38 | | 52.62 |
| | $NH_3$ | 31.3 | 2.48 | |
| 2015 | $NO_3^-$ | | 0.005 | 43.34 |
| | $SO_4^{2-}$ | 11.27 | | 58.37 |
| | $NH_3$ | 32.83 | 2.69 | |
| 2016 | $NO_3^-$ | | 0.004 | 46.27 |
| | $SO_4^{2-}$ | 5.55 | | 45.15 |
| | $NH_3$ | 38.24 | 3.09 | |
| 2017 | $NO_3^-$ | | 0.001 | 58.22 |
| | $SO_4^{2-}$ | 2.56 | | 43.64 |
| | $NH_3$ | 21.88 | 1.23 | |
| Spring | $NO_3^-$ | | 0.001 | 40.55 |
| | $SO_4^{2-}$ | 3.5 | | 49.72 |
| | $NH_3$ | 26.57 | 2.49 | |
| Summer | $NO_3^-$ | | 0.002 | 34.69 |
| | $SO_4^{2-}$ | 27.85 | | 86.23 |
| | $NH_3$ | 58.29 | 1.74 | |
| Autumn | $NO_3^-$ | | 0.002 | 47.34 |
| | $SO_4^{2-}$ | 2.71 | | 49.18 |
| | $NH_3$ | 32.3 | 1.87 | |
| Winter | $NO_3^-$ | | 0.005 | 35.94 |
| | $SO_4^{2-}$ | 6.13 | | 36.08 |
| | $NH_3$ | 26.76 | 1.56 | |

Coefficient of variance: Standard deviation/Mean value*100



### 3.5 Characteristics of local emissions and regional transport

### 3.5.1 Local emissions

The concentration of pollutants is obviously affected by meteorological conditions; for example, WS and WD can affect the accumulation and removal of pollutants (Li et al., 2016). Figs. S13-15 show the annual variation characteristics of NSA and gas precursors affected by the WS and WD using CPF. Overall, the higher WS was accompanied by a lower pollutant concentration. As the WS decreases, the pollution becomes serious, and the pollution hot spots were gradually concentrated. On the whole, when the WS was usually greater than 2 m/s, the pollution was light (pollutant concentration percentile was between 0-25). When WS was usually less than 1 m/s, the pollution was heavy (pollutant concentration percentile was between 75-100), the which also reflects the distance and orientation between the emission source and the observation station, indicating that when the pollution was serious, the contribution of local source emissions was more prominent.

Nitrate and NOx have similar distributions of pollution hot spots in the polar plot diagram (Fig. S13), and when the concentration percentile was between 0-25, they were concentrated in the northeast and southeast directions and widely distributed. When the concentration percentile was between 25-75, the sources of nitrate and NOx were distributed west and northeast of the observatory, and there were important contribution sources in the northwest direction (WS was approximately 3-4 m/s) in 2017. When the WS was approximately 1-2 m/s and the concentration percentile was between 50-75, the important NOx source was in the northwest direction. When the accumulation of pollution concentration was high (concentration percentile was between 75-100), the nitrate source was mainly concentrated in the east and southeast of the observation station, and NOx was distributed in the south and southeast, with WSs of less than 1 m/s; additionally, the distribution of pollution hot spots was relatively wide in 2016 (the annual mean values of NOx were 42.15, 43.99 and 39.63 (ppb) in 2015, 2016 and 2017, respectively). The sulfate and $SO_2$ pollution sources affected by meteorological



conditions also have similar distribution characteristics (Fig. S14). At a higher
concentration of pollutants, the pollution hot spots of sulfate were distributed in the east
and southeast of the observation station, and $SO_2$ was distributed in the northeast,
southeast and west. The concentrations of $SO_2$ were 5.44, 4.15 and 3.68 (ppb) in 2015,
2016 and 2017, respectively. Compared with 2017 and 2016, the distribution of $SO_2$
pollution sources in 2016 was also more extensive, mainly in the west and northeast.
The $NH_3$ emissions were slightly different from those of $SO_2$ and NOx (Fig. S15).
Under conditions of high pollution concentration (concentration percentile was
between 75-100), the pollution hot spots were distributed in the west in 2015 (WS was
approximately 2-3 m/s), in the north in 2016 (WS was approximately 3 m/s), and in the
near distance in 2017 (WS was approximately 0.5 m/s). The higher pollution
concentration was accompanied by a higher WS (2015 and 2016), which indicates that
the $NH_3$ emission transport in the surrounding area was more obvious, which may come
from the surrounding agricultural source distribution area (Liu et al., 2019b;Liu et al.,
2013a). The annual mean value of $NH_3$ emissions in 2017 was 27.91 ppb, which is
significantly higher than those in 2015 and 2016 at 17.93 ppb and 16.55 ppb,
respectively. During the 25-50 concentration percentile period of the $NH_3$, there was a
WS of approximately 2 m/s east of the observation site, and during the 50-75
concentration percentile period, there was an obvious source northwest of the
observation site, with a WS of approximately 4 m/s. During the 75-100 concentration
percentile periods, the pollution sources were mainly local. This shows that in 2017, in
addition to the pollution sources being distributed in the east and southeast, the higher
$NH_3$ emissions were also contributed by the surrounding emission sources northwest of
Chengdu.
**3.5.2 Gaseous precursors of NSA regional transport**
The PSCF is used to analyse the potential source distribution of pollutants to determine
the regional transport characteristics of pollutants (Ji et al., 2019). Fig. 9 shows the
PSCF analysis of NOx, $SO_2$ and $NH_3$, with significant differences in their potential



source distributions. The higher PSCF value of NOx was mainly distributed within 300
km west and southwest of Chengdu in 2015 (Ya'an, Meishan, Leshan and western
Chengdu), north and south of Chengdu in 2016 (Deyang, Meishan, Leshan and
northwestern Chengdu), and south and northeast of Chengdu in 2017 (Deyang,
Mianyang, Meishan, Leshan and western Chengdu). Chengdu is located along the
western margin of the Sichuan Basin. It was also observed through satellite remote
sensing data that the higher $NO_2$ emissions in the Sichuan Basin are distributed in
Chengdu and Chongqing (Fig. S16). As shown in Fig. 9, the higher PSCF values were
concentrated in the surrounding areas of Chengdu, indicating that the Chengdu NOx
was mainly from local emissions. The $SO_2$ emissions were widely distributed, mainly
in the Sichuan Basin. Among them, Leshan city and Meishan city south of Chengdu
have higher $SO_2$ emissions, and another higher emission source was distributed in
Chongqing (Fig. S16). The PSCF analysis of $SO_2$ shows that the higher PSCF values
were distributed in the western, southern and southwestern parts of Chengdu, and the
western, southern and southwestern marginal regions of the Sichuan Basin were also
important potential distribution areas. Therefore, comparison Figs. 9 and S16 shows
that the main source of $SO_2$ may be distributed in the western, southern and
southwestern edge areas of the Sichuan Basin. In particular, Leshan, Ya'an and Meishan
were important potential sources. There were different sources of $NH_3$ emissions from
2015 to 2017, mainly distributed in the Sichuan Province. In 2015, the potential sources
were mainly west of the Sichuan Basin, southwest of Chengdu city, which is
approximately 100 km away from an important source. A higher PSCF was mainly
distributed in Ya'an, Leshan, Meishan and Yibin in 2016. In 2017, the higher PSCF was
mainly distributed in the western and northern areas of the Sichuan Basin, as well as
Meishan and Leshan, which were close to Chengdu and contributed significantly.
Northwest of Chengdu, Deyang, Mianyang and Guangyuan were important potential
sources, and a small part also comes from south of Gansu and Shaanxi. Fig. S17 shows
the Multiresolution Emission Inventory for China (MEIC) Gridded emissions of $NH_3$



in 2016 (Zhang et al., 2019). The higher NH₃ emissions were mainly concentrated in
the interior of the Sichuan Basin, especially near Chengdu, the western edge of the
basin. In comparison with Figs. 9 and S17, the regions with potential impacts on NH₃
concentration in Chengdu were mainly distributed in the Sichuan Basin, especially
south and northeast of Chengdu.

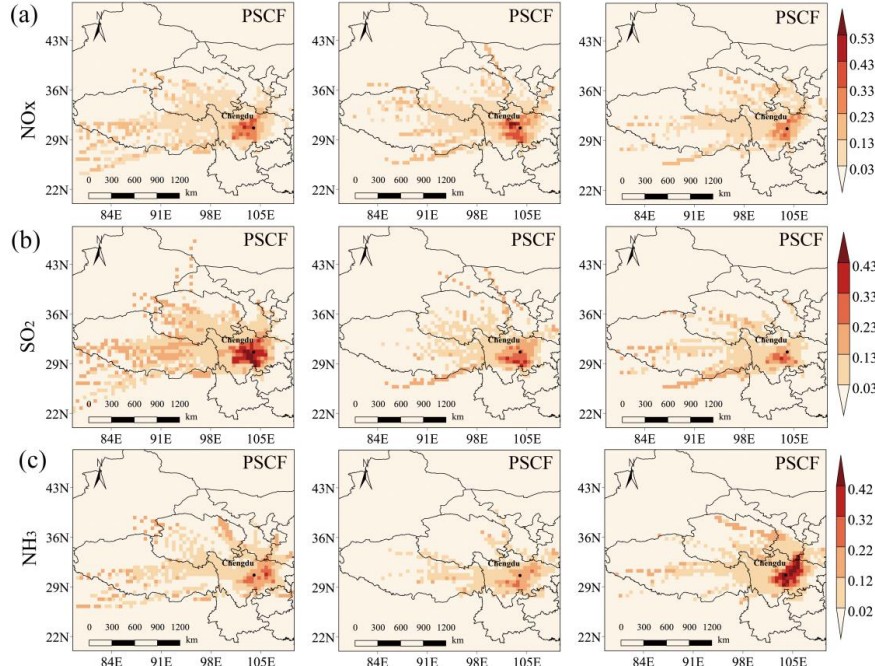


Fig. 9. PSCF of NOx, SO₂ and NH₃ (ppb) in Chengdu from 2015 to 2017.
**4 Conclusions**
The long-term observation experiment with hourly resolution of NSA from January 1,
2015 to December 31, 2017 was carried out in Chengdu in southwest China, which is
in the Sichuan Basin. The pollution characteristics of NSA's annual, monthly, seasonal,
diurnal and weekly variations were demonstrated. The characteristics of chemical
conversion, the relationship with carbonaceous aerosols, and the sensitivity of emission
reduction control were analysed. Finally, combined with meteorological factors and
PSCF simulation, the local emission and regional transport characteristics of NSA



gaseous precursors were also illustrated. The main conclusions were as follows:
(1) Compared with 2015, the concentration of $NO_3^-$ in 2017 did not decrease
significantly, while the concentrations of $SO_4^{2-}$ and $NH_4^+$ decreased. With the increase
in $PM_{2.5}$ concentration, the NSA mass concentration increased, accounting for 31.45-
37.78% of $PM_{2.5}$, but there was a downward trend, indicating that the contribution of
other unknown components to $PM_{2.5}$ may significantly increase with the aggravation of
pollution. Higher and lower NSA concentrations were seen in winter and summer,
respectively, and higher concentrations were seen more during the day than at night.
Although the NSA concentration on weekdays was slightly higher than that on
weekends, the mean difference between them was nonsignificant.
(2) With the increase in $PM_{2.5}$ concentration, aqueous phase oxidation was an important
process of NSA chemical conversion. The ammonia-rich environment in Chengdu
became increasingly obvious. Under this condition, the main strategy to reduce the
concentration of NSA was to continue to promote sulfate reduction and to strengthen
the control of nitrate and ammonium reductions. When controlling the decrease in
sulfate and nitrate, the decrease in ammonium will be obvious. NSA synergistic
emissions reduction control implementation can achieve a better emission reduction
effect. Regulation of the emission reduction ratio of NSA and reduction of the impact
on aerosol pH was also a problem worth further consideration.
(3) Local emissions and regional transport of NSA gaseous precursors have an
important impact on air pollution in Chengdu. In particular, NOx was the most obvious
contribution from the western, southern and southwestern margins of the Sichuan Basin
and local emissions in Chengdu. Northeast and west of Chengdu, there were high local
$SO_2$ emission sources, and combined with satellite remote sensing data and PSCF
analyses, within the Sichuan Basin, the cities of Leshan and Meishan south of Chengdu
may be important sources of $SO_2$ regional transport. The potential sources of $NH_3$ were
widely distributed, and the internal emissions of the Sichuan Basin may be important
potential contribution sources. Southwest, south and southeast of Chengdu, the





contribution was obvious. The analysis of local emissions and regional transport shows
that implementing regional joint prevention, controlling emissions reduction working
mechanisms and simultaneously promoting pollutant emission control are important
implementation plans.
**Acknowledgements**
This work was supported by the People's Republic of China Science and Technology
Department (No. 2018YFC0214001 and No. 2016YFC0202000) and the National
Natural Science Foundation of China (No. 91544221).
**Data availability**
The data are available on request to the corresponding author.
**Author contribution**
XL, QT and LK designed and led this study. QT and MF was responsible for the
observations. LK, MF, YL, YZ, CZ, CL analyzed the data. LK, YQ, JA, NC, YD, RZ
and ZW discussed the results. LK and XL wrote the paper. All authors commented on
the paper.
**Competing interests**
The authors declare that they have no conflict of interest.

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
