# Peer review of "Elucidating the pollution characteristics of nitrate, sulfate and ammonium in"

_Atmospheric Chemistry and Physics, 2019_

## Referee Comment (RC1) · Anonymous Referee #1 · 19 Mar 2020

This study provided a good data basis for explaining the pollution characteristics of secondary inorganic aerosols in PM2.5 through a long-term atmospheric observation experiment. The formation mechanism and role of secondary inorganic aerosols during the formation of haze are still a research hot issue. The author not only explained the pollution characteristics of nitrate, sulfate and ammonium in Chengdu, but also analyzed its formation mechanism through observation data and ISORROPIA-II thermodynamic model simulation. Finally, the author also analyzed the distribution characteristics of pollution emissions and potential sources in Chengdu. At present, China is strengthening the control and treatment of air pollution, a long-term atmospheric observation experiment has a high research value for the analysis of the formation of air

pollution and the implementation of haze abatement measures. However, there are some writing, grammatical and technical errors in the paper, and it is suggested that the author carefully revise and organize the presentation of the paper.

Line. 53-54, PM2.5 interpretation is inaccurate. "PM2.5(aerodynamic diameter less than 2.5 $\mu$m)"

Line. 58-61, "NSA" is the abbreviation of nitrate, sulfate and ammonium in the paper? please rewrite this sentence.

Line. 77-78, suggest reinterpreting this sentence, how to understand the "regional transport"

Line. 82-83, if the author defines an abbreviation at the beginning of the paper, it is recommended to use the abbreviation below. Please use "NSA" abbreviations instead of nitrates, sulfates and ammonium.

Line. 87, please correct this writing, "(2013-207)"

Line. 106-109, "high time resolution", what's the meaning of this?

Line. 141, in Table 1, parameters not covered in this paper can be removed, such as PM1 and H2S

Line. 185-186, percentile (e.g. 0-25, 25-50, 50-75 and 75-100), please confirm it is consistent with the title of Fig. 13-15 (0-25%, 25-50%, 50-75%, and 75-100%.) in Supplementary materials.

Line. 185-186, it is suggested that the author briefly describe what measures should be taken.

Line. 263, "These variations have similar trends due to meteorological factors", what do you mean?

Line. 306, the legend in Fig. 3 is suggested to be modified, with one reserved, and

also pay attention to modify other pictures, such as Fig.S2 and S8.

Line. 309, replace "daily changes" with "diurnal variation".

Line. 400, please explain what "r" in Fig. 5 means?

Line. 452, in Fig.7h, "SO42- gas-particle phase partitioning"? inconsistent with the NH4+ in the picture.

Line. 470, please explain what "k" in Fig. 8 means?

In Section 3.5.2, authors are advised to supplement PSCF analysis of NO2 and NO.

Line. 470, modify the Fig.9, remove the repeat "PSCF" in the picture

Line. 521-525, the description is too simple, please rewrite the research results.

Authors are requested to write rules uniformly. It is not recommended to use abbreviations in the title of the figures, such as NSA, AWC, SOR, NOR PSCF. In addition, there are "(a)", "(b)" and "(c)" in the picture, please explain what it means in the title.

In Fig. 5 and Fig. S4, please confirm the carbon monoxide (CO) unit, "ppb" and "ppm"?

In Supplementary materials, Line. 90 and 96, pay attention to writing, it shouldn't be "2107"

---

## Referee Comment (RC2) · Anonymous Referee #2 · 8 May 2020

Comments on "Elucidating the pollution characteristics of 1 nitrate, sulfate and ammonium in PM2.5 in Chengdu, southwest China based on long-term observations" by Kong et al.

This paper presented an overview of air pollution in Chengdu, southwest China based on study, a three-year observations of gas and particulate pollutants. Probably due to the special topography, the data from this site shows special characteristics of pollutants, different from most other polluted regions in China, e.g., North China plain and Pearl River Delta etc. Thermodynamic models and trajectory analysis have also been applied to analyze aerosol pH, partitioning of inorganic semi-volatile species and the

contribution of potential source regions. Overall, I think that the datasets are very interesting and valuable, and authors have tried to give a comprehensive overview and analysis of the mechanisms beyond. However, there is still much room for improvement. I would suggest the authors to carefully consider my comments/suggestions in their revision before its final publication in ACP.

Major concern: 1. Meteorological parameters and PM compositions show distinct diurnal variations compared to other regions such as NCP, PRD and YRD. Though chemistry is in this game, I guess that the special topography may play a dominant role in these features, which is missing from this submission. I'd suggest the authors to add these discussions. 2. QA/QC. Quality assurance and control is essential for multi-year analysis. Maybe I overlooked it and I didn't find a description to assure the data quality. QA/QC would give the readers more confidence in your data and analysis, e.g., the extremely high NH3 in the winter of 2017. 3. To avoid jump between text and SI, the authors could consider moving some SI parts into the main text. For example, gas phase NH3 that frequently used and discussed is missing from the main text.

Minor comments: Abstract: Line 21 "a long-term observational experiment was conducted from January 1, 2015 to December 31, 2017" Three years measurements are longer than a campaign-based experiment but I won't call it "long-term".

Line 27 "Seasonal and diurnal variations have obvious characteristics, winter still has a high NSA concentration and emission intensity, and the concentration during the day was higher than that at night. " This is unusual; is it because of the valley topography?

Line 34 "The ammonia-rich environment became increasingly obvious in the atmosphere of Chengdu" It is not clear what you want to say. Do you mean that you see an increase in NH3 concentration or partition rate?

Page 3 line 69 "For example, photochemistry may affect the formation of NSA at high solar radiation, and the homogeneous reaction may dominate the formation of NSA in high relative humidity" I think you mean "heterogeneous" instead of homogeneous ?

[Figure]

Page 3 line 82 "The characteristics of higher concentrations proportion of nitrate, sulfate and ammonium in PM2.5 were also found in other polluted areas in China, such as Beijing-Tianjin-Hebei, the Yangtze River Delta, the Pearl River Delta, the Fenwei Plain, " You may not use "other" since you also refer to Beijing Tianjin-Hebei.

Page 5 line 123, Sect 2.1 According to the high NO concentrations, the site could be quite close to adjacent sources. This should be mentioned in the site description.

Page 7 line 154 "Temperature (T), relative humidity (RH) and the total concentrations (i.e., gas + aerosol) of Na+, SO42−, NH3, NO3 −, Cl-, 155 Ca2+, K+ and Mg2+ were input into the ISORROPIA-II thermodynamic mode" Do you have HCl, and HNO3 measured? I don't see it in your instrument list. You may need to do a back calculation to check the modelled value and see if you may retrieve these information iterative model calculations. You may need to calculate the uncertainties or bias due to these missing data in your model input.

Page 7 line 159 "The simulated data and observed data were compared and analysed. Simultaneously, the aerosol water content (AWC) and pH of aerosols were calculated. The sensitivity of the interaction between aerosol chemical components (NSA) was analysed (Ding et al., 161 2019;Fountoukis et al., 2009). Could you show a comparison between the modelled and measured gas phase NH3, HCl and HNO3? This result can be used to check the reliability and performance of thermodynamic models.

Page 8 line 178 "the conditional probability function (CPF) was introduced the R Programming Language." Complete the sentence.

Figure 2, why both fractional contribution of both organic and inorganic decrease at high PM2.5 concentrations? What's the other compositions that are increasing?

Page 9 line 216 "The annual average mass concentration of NSA also changed significantly, and the difference was large. The Mann-Whitney U test showed that the variation in NO3 - was nonsignificant (p > 0.05), and SO4 2- and NH4 + had obvious

significance from 2015 to 2017 (p < 0.05), indicating that NO3 - had not decreased significantly, and there was an increase in 2017 compared to 2015." Here, you could further discuss the reasons why the concentration SO2 and sulfate decease more than that of NOx and nitrate.

Page 12, line 283 " This also shows that the implementation of air pollution reduction measures should increase the emission reduction intensity in terms of NOx and NH3 emissions, especially the implementation of autumn and winter air pollution prevention and control action." You were talking about high NH3 in 2017 and then taking to NOx? I am missing a link here. Also I'd like to see an explanation about the high NH3 concentration up to 60 ppb. This is very high for a monthly average. What's the pH under this condition?

Page 13 line 293 "Sulfate has a significant downward trend in all seasons from 2015 to 2017, especially in winter. This downward trend was due to implementation of the Air Pollution 294 Prevention and Control Action Plan" Such discussion should be put in Sect 3.1.

Page 15 line 325 "As shown in Fig. S4, from 9:00 to 11:00 a.m., the concentrations of SO2, NOx, NH3, CO and other gases increased significantly, indicating that the primary emission of pollutants was relatively strong. At this time, higher RH (Fig S5) also provides favourable conditions for the formation of secondary aerosols and promotes the accumulation of NSA" But the RH in Fig. S5 is decreasing in contrast to an increase in aerosol concentrations?

Sect. 3.4.1 In general, it is true that the emissions of multi-pollutant may come from the same kinds of sources. But you cannot draw such a conclusion based on correlation studies. Because the variation of most pollutants, especially those of long-lifetime, is strongly influenced by the boundary layer developments, and may show a similar diurnal variation in spite of different origins (sources).

Page 20 Line 430 "Figure 7 shows the variation characteristics of NSA chemical conversions and meteorological conditions with increasing RH. SOR and NOR increased with increasing RH, suggesting that SO2 and NO2 were more likely to produce sulfate and nitrate under higher RH conditions. In Fig. 7, how did you do the calculation, classifying the data according to RH or you keep all input the same but change RH only? In the former case, the apparent correlation with RH may not represent the real causation as chemical compositions and other parameters may change also change.

Sect 3.5.2 I understand that the authors adopted this approach based on a published study. This approach, however, is subject to several problems, e.g., neglecting the dilution of pollution in the course of transport which may overestimate the contribution of distant sources, or the endpoint is not necessary at the ground level, or why 24 hour (aerosols have a longer lifetime) etc. If you still want to keep this part, please explicitly include these caveats in the text to avoid misinterpretation of this result.

---

## Author Comment (AC1) · 18 Jun 2020

Dear Reviewer,

We would like to thank you for your great effort and detailed work on this manuscript. We have revised the manuscript and responded to each of the comments from the reviewers. In our response, your questions are shown in italics, and the responses are shown in standard text. For the ACP discussion, our research team also performed further analysis of the research results and made minor modifications to this manuscript.

We appreciate your help and time.

[Figure]

Sincerely yours, Xingang Liu and coauthors. School of Environment Beijing Normal University 100875 Beijing China E-mail: liuxingang@bnu.edu.cn; lxgstar@126.com Tel: +86-13810193569

——————————————————————————————————— Title: Elucidating the pollution characteristics of nitrate, sulfate and ammonium in PM2.5 in Chengdu, southwest China, based on three-year measurements ————————— ——————————————————————————————————— Response to Anonymous Referee This study provided a good data basis for explaining the pollution characteristics of secondary inorganic aerosols in PM2.5 through a long-term atmospheric observation experiment. The formation mechanism and role of secondary inorganic aerosols during the formation of haze are still a research hot issue. The author not only explained the pollution characteristics of nitrate, sulfate and ammonium in Chengdu, but also analyzed its formation mechanism through observation data and ISORROPIA-II thermodynamic model simulation. Finally, the author also analyzed the distribution characteristics of pollution emissions and potential sources in Chengdu. At present, China is strengthening the control and treatment of air pollution, a long-term atmospheric observation experiment has a high research value for the analysis of the formation of air pollution and the implementation of haze abatement measures. However, there are some writing, grammatical and technical errors in the paper, and it is suggested that the author carefully revise and organize the presentation of the paper. Response: We appreciate your comments and have revised the full text; we also regret each error and have discussed and corrected them. ——————————————— ——————————————————————————————————— Line. 53-54, PM2.5 interpretation is inaccurate. "PM2.5(aerodynamic diameter less than 2.5 $\mu$m)" Response: We appreciate your comments and apologize for our unprofessional description. Now, it reads as follows: particles with aerodynamic equivalent diameter $\leq$ 2.5 $\mu$m in ambient air (PM2.5, also known as fine particles) ——————————————————————— ——————————————————————— Line. 58-61, "NSA" is the abbreviation of nitrate, sulfate and ammonium in the paper? please rewrite this sentence. Response: We

appreciate your comments and apologize for our unprofessional description. Now, it reads as follows: Nitrate, sulfate, ammonium, organic matter and elemental carbon are the main components of PM2.5, among which nitrate, sulfate, and ammonium (NSA) are the main secondary inorganic aerosols in PM2.5. ————————————————————————————————————————————- Line. 77-78, suggest reinterpreting this sentence, how to understand the "regional transport" Response: We appreciate your comments and apologize for our unprofessional description. Now, it reads as follows: In addition to the air pollution caused by the local emission of pollutants, the regional transportation of pollutants from its surrounding cities also has an important impact on the urban air quality. ————————————————————————————————————————- Line. 82-83, if the author defines an abbreviation at the beginning of the paper, it is recommended to use the abbreviation below. Please use "NSA" abbreviations instead of nitrates, sulfates and ammonium. Response: We appreciate your comments and have revised the manuscript. Now, it reads as follows: Higher concentrations of NSA in PM2.5 were also found in regions with more serious air pollution in China, such as Beijing-Tianjin-Hebei, the Yangtze River Delta, the Pearl River Delta, the Fenwei Plain, and the Chengdu-Chongqing region. ————————————————————————————————————————————

Line. 87, please correct this writing, "(2013-207)" Response: We appreciate your comments and have revised the text. Now, it reads as follows: In response to this situation, the Chinese government issued an Air Pollution Prevention and Control Action Plan (2013-2017) in 2013 to reduce pollutant emissions and improve air quality. ————————————————————————————————————————- Line. 106-109, "high time resolution", what's the meaning of this? Response: We appreciate your comments and apologize for our unprofessional description. We mean that the time interval for gathering observation data is relatively small, at 1 hour. Compared with the daily average data of manual operation sampling, the time resolution is higher; we have revised and polished this sentence. Now, it reads as follows: However, these analyses may be affected by the experimental equipment, observation stations and

other conditions, and the time span of these atmospheric observations usually includes several pollution processes or lasts for weeks or months. Thus, it is difficult to analyse the long-term variations in characteristics of air pollution through comprehensive observation. In particular, there are few high-time-resolution (1 hour) observation experiments carried out with online automatic observation systems. ———————————————————————————————————— Line. 141, in Table 1, parameters not covered in this paper can be removed, such as PM1 and H2S Response: We appreciate your comments and have revised it. Now, it reads as follows:   Table 1. The experimental instruments used in this study Instruments Parameters Manufacturer/Country URG-9000 NO3-/SO42-/NH4+/Na+/Mg2+/Ca2+/Cl-/K+ Thermo Fisher Scientific/USA SHARP 5030 PM2.5 Thermo Fisher Scientific/USA RT-4 OC/EC Sunset Laboratory/USA Xact-625 Metal elements Cooper Environmental Services /USA 17i/450i/48i/49i NOx/NO2/NO/NH3/SO2/CO/O3 Thermo Fisher Scientific/USA WXT520 Meteorological parameters VAISALA/Germany OC: organic carbon; EC: element carbon ———————————————————————————————————— Line. 185-186, percentile (e.g. 0-25, 25-50, 50-75 and 75-100), please confirm it is consistent with the title of Fig. 13-15 (0-25%, 25-50%, 50-75%, and 75-100%.) in Supplementary materials. Response: We appreciate your comments and have revised the text in the Supplementary Materials. We have made the writing consistent throughout the manuscript and adopted percentiles (e.g., 0-25, 25-50, 50-75 and 75-100), and the "%" in the figure title was deleted. ———————————————————————————————————— Line. 185-186, it is suggested that the author briefly describe what measures should be taken. Response: We appreciate your comments. There is no description of "measures" in line 185, and we presume that the reviewer refers to the description in lines 228-230 about the measures taken in the Air Pollution Prevention and Control Action Plan. We also describe this part in detail. Now, it reads as follows: From 2015 to 2017, the measures taken by Sichuan Province in the coordinated reduction of multiple pollutants have been continuously strengthened, and the scope of management and control has been

continuously expanded, for example, in the improvement of desulfurization, denitrification and dust removal technologies in key industries, from accelerated improvement in 2015 to deeper improvement in 2017. The process of eliminating small coal-fired boilers began in 2015 and was completed in 2017, when the ultra-low-emission coal-fired power plant transformation was promoted. In terms of vehicle emission control, we accelerated the elimination of "yellow label" vehicles (general term for gasoline vehicles with emission levels lower than the national I emission standard and diesel vehicles with emission levels lower than the national III emission standard when new vehicles are finalized) and "old vehicles" (the emission level does not meet the national stage IV emission standard) in 2015 and basically completed the elimination of yellow standard vehicles in 2017. The quality supervision of oil products has also been improved, and non-road mobile machinery pollution control requirements were proposed in the 2017 plan (ThePeople'sGovernmentofSichuanProvince, 2015, 2016, 2017). References: The People's Government of Sichuan Province. Detailed rules for the implementation of the action plan for the prevention and control of air pollution in Sichuan Province 2015 annual implementation plan. WebsiteïijŽ http://www.sc.gov.cn/10462/10883/11066/2015/4/22/10333390.shtmlïijŇlast access: June 17 2020. The People's Government of Sichuan Province. Detailed rules for the implementation of the action plan for the prevention and control of air pollution in Sichuan Province 2016 annual implementation plan. WebsiteïijŽ http://www.sc.gov.cn/zcwj/xxgk/NewT.aspx?i=20160401095908-612769-00-000ïijŇlast access: June 17 2020. The People's Government of Sichuan Province. Detailed rules for the implementation of the action plan for the prevention and control of air pollution in Sichuan Province 2017 annual implementation plan. WebsiteïijŽ http://www.sc.gov.cn/zcwj/xxgk/NewT.aspx?i=20170527091543-450025-00-000ïijŇlast access: June 17 2020. ————————————————

———————————————————- Line. 263, "These variations have similar trends due to meteorological factors", what do you mean? Response: We appreciate your comments and apologize for our unprofessional description. What we want to say is

that the meteorological conditions also have obvious monthly variation characteristics, which may have some influence on the variation characteristics of NSA. Now, the text reads as follows: The monthly variation characteristics of NSA from 2015 to 2017 are shown in Fig. 3. At the beginning and end of each year, the pollutant concentration is relatively high and relatively low in the middle of each year. The meteorological conditions also have obvious monthly variation characteristics (Fig. S5 a and b); from April to August, they have higher WS and lower RH, which is not only conducive to the dilution and diffusion of pollutants but also reduces the chemical conversions of pollutants by aqueous phase and influences the formation of secondary inorganic aerosols. ————————————————————————————————————

—- Line. 306, the legend in Fig. 3 is suggested to be modified, with one reserved, and also pay attention to modify other pictures, such as Fig.S2 and S8. Response: We appreciate your comments and apologize for our unprofessional description. We reviewed similar problems in the other figures and have corrected them. —————————

——————————————————————————————————- Line. 309, replace "daily changes" with "diurnal variation". Response: We appreciate your comments and have revised the text. ———————————————————————————

————————————- Line. 400, please explain what "r" in Fig. 5 means? Response: We appreciate your comments and apologize for our unprofessional description. This text has been revised in the manuscript, and this error will not appear in the new manuscript. ———————————————————————————————

——— Line. 452, in Fig.7h, "SO42- gas-particle phase partitioning"? inconsistent with the NH4+ in the picture. Response: We appreciate your comments and apologize for our unprofessional description. The analysis content has been modified in this section, and the comments made by you in the manuscript have also been resolved. —————————————————————————————————————————- Line. 470, please explain what "k" in Fig. 8 means? Response: We appreciate your comments and apologize for our unprofessional description. Now, the text reads as follows: Fig. 8. Molar ratio analysis of NSA (nitrate, sulfate and ammonium). (a)

Interannual variation in the molar ratio of SO42- and NH4+. (b) Interannual variation in the molar ratio of NO3- and NH4+. (c) Seasonal variation in the molar ratio of SO42- and NH4+. (d) Seasonal variation in the molar ratio of NO3- and NH4+. k: Fitting slope of linear regression. ————————————————————————————————————

————————— In Section 3.5.2, authors are advised to supplement PSCF analysis of NO2 and NO. Response: We appreciate your comments and have revised the text. ————————————————————————————————————————————-

Line. 470, modify the Fig.9, remove the repeat "PSCF" in the picture Response: We appreciate your comments and apologize for our unprofessional description. This issue is also associated with the previous issue, and we have solved the problem. ———————————————————————————————————————————- Line.

521-525, the description is too simple, please rewrite the research results. Response: We appreciate your comments and apologize for our unprofessional description. Now, the text reads as follows: The results of using the ISORROPIA-II thermodynamic equilibrium model to simulate NO3-, SO42- and TNH3 emission reduction control effects of 5%, 10%, 15% and 20%, respectively, are shown in Table S3, showing that controlling the concentration of NO3- and SO42- is also helpful to reduce the concentration of NH4+ and indicating that controlling its precursor NOx and SO2 is of great significance to reduce the secondary inorganic aerosol in PM2.5 (the detailed results are described in the supplementary materials). supplementary materials Through observation data quality control, we screened 618 sample input ISORROPIA-II thermodynamic equilibrium models to ensure the integrity of the samples and the effectiveness of the data. The control variable method was used to explore the impact of a concentration reduction for other species. For example, to explore the impact of NO3- concentration reduction for SO42- and NH4+, the NO3- data were calculated based on the 5, 10, 15 and 20% emission reduction ratio, and other parameters were input into the model using the observation data to explore the relative variable of SO42- and NH4+ concentration. The simulation results are shown in Table S3. When only NO3- and SO42- were reduced, NH4+ was significantly reduced, but the changes

in SO42- and NO3- were not obvious, and the relative variables of approximately 12% and 7% may be mainly affected by the change in phase state. When only TNH3 was controlled, the relative variable of SO42- was not obvious, and the concentrations of NO3- and NH4+ decreased, but the relative variable was not large. NSA has a good reduction effect under synergistic emission reduction control. The results show that reducing the amount of NO3- and SO42- can not only reduce their concentrations but also help to reduce the concentration of NH4+. It also suggests that controlling the gaseous precursors NOx and SO2 is of great significance to reduce the amount of secondary inorganic aerosol in PM2.5. Studies in Mexico City have also shown that reducing total sulfates and total nitrates rather than total ammonium helps reduce PM2.5 concentrations in an ammonium-rich environment (Fountoukis et al., 2009). Table S3. Simulation of NO3-, SO42- and TNH3 emission reduction control effect (%) and its influence on pH based on the ISORROPIA-II thermodynamic model. Reduction Only NO3- Reduction Only SO42- Reduction NO3- SO42-* NH4+ pH* NO3- SO42-* NH4+ pH 5% 11.92 12.25914 8.33 4.0495 7.19 17.1088 9.77 4.08 10% 16.58 12.25911 11.13 4.0519 7.09 21.9593 13.65 4.13 15% 21.23 12.25909 13.91 4.0546 7.00 26.8093 17.50 4.19 20% 25.88 12.25906 16.69 4.0547 6.91 31.6596 21.58 4.25 Only TNH3 Reduction Synergistic ** NO3- SO42-* NH4+ pH NO3- SO42-* NH4+ pH 5% 7.51 12.25938 5.85 4.02 12.08 17.1090 12.86 4.07 10% 7.79 12.25965 6.20 3.99 16.85 21.9596 19.80 4.09 15% 8.10 12.25998 6.59 3.95 21.64 26.8097 26.17 4.11 20% 8.45 12.26040 7.03 3.91 26.37 31.6601 33.29 4.16 Notes: NO3-, SO42- and TNH3 are the concentration variables relative to the observation data; pH is the average; TNH3: NH3 + NH4+; *: In order to display the data difference, the number of digits after the decimal point was increased **: NO3-, SO42- and TNH3 decreased in the same proportion Reference: Fountoukis, C., Nenes, A., Sullivan, A., Weber, R., Van Reken, T., Fischer, M., Matias, E., Moya, M., Farmer, D., and Cohen, R. C.: Thermodynamic characterization of Mexico City aerosol during MILAGRO 2006, Atmospheric Chemistry and Physics, 9, 2141-2156, 10.5194/acp-9-2141-2009, 2009.

Authors are requested to write rules uniformly. It is not recommended to use abbreviations in the title of the figures, such as NSA, AWC, SOR, NOR PSCF. In addition, there are "(a)", "(b)" and "(c)" in the picture, please explain what it means in the title. Response: We appreciate your comments and apologize for our unprofessional description. We have checked and corrected the abbreviations for consistency throughout the manuscript. ——————————————————————————————
——————————————- In Fig. 5 and Fig. S4, please confirm the carbon monoxide (CO) unit, "ppb" and "ppm"? Response: We appreciate your comments and apologize for our unprofessional description. The unit of carbon monoxide (CO) is ppm, and we have revised it. ————————————————————————————————
—————- In Supplementary materials, Line. 90 and 96, pay attention to writing, it shouldn't be "2107" Response: We appreciate your comments and have revised it. ————————————————————————————————————————-

Please also note the supplement to this comment:
https://www.atmos-chem-phys-discuss.net/acp-2019-1142/acp-2019-1142-AC1-supplement.pdf

---

## Author Comment (AC2) · 18 Jun 2020

Response to Reviewer Comments

Dear Reviewer,

We would like to thank you for your great effort and detailed work on this manuscript. We have revised the manuscript and responded to each of the comments from the reviewers. In our response, your questions are shown in italics, and the responses are shown in standard text. For the ACP discussion, our research team also performed further analysis of the research results and made minor modifications to this manuscript.

[Figure]

We appreciate your help and time.

Sincerely yours, Xingang Liu and coauthors. School of Environment Beijing Normal University 100875 Beijing China E-mail: liuxingang@bnu.edu.cn; lxgstar@126.com Tel: +86-13810193569

——————————————————————————————————————- Title: Elucidating the pollution characteristics of nitrate, sulfate and ammonium in PM2.5 in Chengdu, southwest China, based on three-year measurements ————————— ————————————————————————————- Response to Anonymous Referee This paper presented an overview of air pollution in Chengdu, southwest China based on study, a three-year observations of gas and particulate pollutants. Probably due to the special topography, the data from this site shows special characteristics of pollutants, different from most other polluted regions in China, e.g., North China plain and Pearl River Delta etc. Thermodynamic models and trajectory analysis have also been applied to analyze aerosol pH, partitioning of inorganic semi-volatile species and the contribution of potential source regions. Overall, I think that the datasets are very interesting and valuable, and authors have tried to give a comprehensive overview and analysis of the mechanisms beyond. However, there is still much room for improvement. I would suggest the authors to carefully consider my comments/suggestions in their revision before its final publication in ACP. Response: We appreciate your comments and have revised the manuscript accordingly. We firmly believe that your guidance is of great significance for improving our research. ——————————————————————————————————————- Major concern: 1. Meteorological parameters and PM compositions show distinct diurnal variations compared to other regions such as NCP, PRD and YRD. Though chemistry is in this game, I guess that the special topography may play a dominant role in these features, which is missing from this submission. I'd suggest the authors to add these discussions. Response: We appreciate your comments and have revised the text. We also noticed that during the data analysis, the pollutant concentration in the daytime

was higher, which was obviously different from that in NCP, PRD and YRD. We also collated the relevant research results in NCP, PRD and YRD and found that they were consistent with the questions you raised. Therefore, according to your guesses and prompts, we conducted an in-depth analysis. The unique topographic structural features did have an important impact on the diurnal variation of meteorological factors in the Sichuan Basin, which may also be an important factor that causes the daily changes in air pollutants with unique characteristics. We also supplement and discuss the corresponding parts of the manuscript. Now, the text reads as follows: In previous studies in Beijing-Tianjin-Hebei and the Pearl River Delta, the concentration of pollutants was affected by meteorological factors, and it was usually lower in the daytime than at night. In the Yangtze River Delta, the peak usually occurs in the morning, but in our study, the concentration was higher in the daytime than at night (Peng et al., 2011;Wang et al., 2018;Guo et al., 2017b). In addition to the diurnal variations in WS and atmospheric humidity, some studies have shown that due to the unique topographical structure of the Sichuan Basin, the atmospheric circulation between the Qinghai-Tibet Plateau, Yunnan-Guizhou Plateau and Sichuan Basin and the meteorological conditions of the Chengdu region are affected, such as the characteristics of air mass transport and typical "night rain" (more precipitation at night than in the day) under the influence of atmospheric circulation (Zhang et al., 2019b;Zhang et al., 2019a). Reference: Guo, J., Xia, F., Zhang, Y., Liu, H., Li, J., Lou, M., He, J., Yan, Y., Wang, F., Min, M., and Zhai, P.: Impact of diurnal variability and meteorological factors on the PM2.5 - AOD relationship: Implications for PM2.5 remote sensing, Environmental pollution, 221, 94-104, 10.1016/j.envpol.2016.11.043, 2017b. Peng, G., Wang, X., Wu, Z., Wang, Z., Yang, L., Zhong, L., and Chen, D.: Characteristics of particulate matter pollution in the Pearl River Delta region, China: an observational-based analysis of two monitoring sites, Journal of Environmental Monitoring, 13, 1927-1934, 10.1039/c0em00776e, 2011. Wang, L., Li, W., Sun, Y., Tao, M., Xin, J., Song, T., Li, X., Zhang, N., Ying, K., and Wang, Y.: PM2.5 Characteristics and Regional Transport Contribution in Five Cities in Southern North China

[Figure]

Plain, During 2013–2015, Atmosphere, 9, 157, 10.3390/atmos9040157, 2018. Zhang, L., Guo, X., Zhao, T., Gong, S., Xu, X., Li, Y., Luo, L., Gui, K., Wang, H., Zheng, Y., and Yin, X.: A modelling study of the terrain effects on haze pollution in the Sichuan Basin, Atmospheric Environment, 196, 77-85, 10.1016/j.atmosenv.2018.10.007, 2019a. Zhang, Y., Xue, M., Zhu, K., and Zhou, B.: What Is the Main Cause of Diurnal Variation and Nocturnal Peak of Summer Precipitation in Sichuan Basin, China? The Key Role of Boundary Layer Low‐Level Jet Inertial Oscillations, Journal of Geophysical Research: Atmospheres, 124, 2643-2664, 10.1029/2018jd029834, 2019b. —————

———————————————————————————————————- 2. QA/QC. Quality assurance and control is essential for multi-year analysis. Maybe I overlooked it and I didn't find a description to assure the data quality. QA/QC would give the readers more confidence in your data and analysis, e.g., the extremely high $NH_3$ in the winter of 2017. Response: We appreciate your comments and apologize for our unprofessional description. We agree with you very much because the instruments involved in this study are online monitoring equipment and have high time resolution (1 hour), so data quality assurance and control are the keys to determining the accuracy and scientific nature of this study. Therefore, we added this information to the supplementary materials. Now, it reads as follows: Data quality control and assurance are important components of atmospheric comprehensive observation experiments. In addition to regular inspection and correction of the equipment through professional operation and maintenance to ensure the accuracy of experimental data, the quality control and processing of monitoring data, such as excluding outliers and data beyond the detection limit, are also an important. As shown in Figs. S1-4, the time sequence of monitoring data and the red part in the figure indicate that the data are missing and that the overall data integrity is good. The missing rates of PM10, PM2.5 and PM1 data in Fig. S1 are 5.0, 6.8 and 6.1%, respectively. The missing rates of $NO_3^-$, $SO_4^{2-}$, $NH_4^+$, OC and EC data in Fig. S2 are 18.3, 17.1, 20.9, 15.2 and 19.6%, respectively. In Fig. S3, the gaseous pollution of NO data is missing 18.2%, $NH_3$ is missing 11.3%, and other gases are missing 9%. The quality of meteorological data is good (Fig. S4),

and the overall missing rate is 3.1% or less. On the whole, the observation data are good and do not affect the continuity of the data as a whole. The Cl-, Na+, K+, Mg2+ and Ca2+ data are significantly missing, and this study only involved in analysis of the ISORROPIA-II thermodynamic equilibrium model. To ensure that each sample data point can be input into the model completely, 618 sample input models are selected according to the data quality control to eliminate the impact of missing ion data to ensure that the model analysis results are effective.

Fig. S1. PM10, PM2.5 and PM1 data quality assurance and control

Fig. S2. NO3-, SO42-, NH4+, OC (organic carbon) and EC (element carbon) data quality assurance and control

Fig. S3. NOx, SO2, NO2, NO, CO and NH3 data quality assurance and control

Fig. S4. Relative humidity (RH), temperature (T), atmospheric pressure (P), wind speed (WS) and wind direction (WD) data quality assurance and control ——————————————————————————————————————————- 3. To avoid jump between text and SI, the authors could consider moving some SI parts into the main text. For example, gas phase NH3 that frequently used and discussed is missing from the main text. Response: We appreciate your comments and apologize for our unprofessional description. We have adjusted the positions of the figures according to the structure of the whole manuscript. ————————————————————————————————————————- Minor comments: Abstract: Line 21 "a long-term observational experiment was conducted from January 1, 2015 to December 31, 2017" Three years measurements are longer than a campaign-based experiment but I won't call it "long-term". Response: We appreciate your comments and apologize for our unprofessional description. We agree with you that the expression "long-term" observation instead of "three-year" observation is not accurate enough, so we use "three-year" observation. ——————————————————————————————————————— Line 27 "Seasonal and diurnal variations have obvious

characteristics, winter still has a high NSA concentration and emission intensity, and the concentration during the day was higher than that at night. "This is unusual; is it because of the valley topography Response: We appreciate your comments and apologize for our unprofessional description. As you commented in "Major concern: 1", we have performed a comparative study of other regions of China, Beijing-Tianjin-Hebei, the Yangtze River Delta and the Pearl River Delta, and found that, as you believe, the diurnal variation of air pollutants in Sichuan Basin is indeed different from that in other regions. Based on related research results, the unique topographical structure of Sichuan Basin will indeed affect the meteorological conditions of Sichuan Basin by affecting the atmospheric circulation (interaction of the Qinghai Tibet Plateau, Yunnan Guizhou Plateau and Sichuan Basin). Therefore, the daily variation characteristics of pollutants may be significantly affected by meteorological conditions. In response to "Major concern: 1", we have made corresponding modifications in the manuscript. ——————————————————————————————————————————- Line 34 "The ammonia-rich environment became increasingly obvious in the atmosphere of Chengdu" It is not clear what you want to say. Do you mean that you see an increase in NH3 concentration or partition rate? Response: We appreciate your comments and apologize for our unprofessional description. We have reconsidered our research purpose and found that the expression of the "ammonia-rich" environment is not accurate. Therefore, we revised the expression based on our current research results. In section 3.4.3 of the manuscript, the ratio of NH4+ to NO3- and SO42- was analysed and found to increase from 2015 to 2017. The results may be attributed to the fact that the concentration of PM2.5 gradually decreased under the implementation of relevant air pollution control measures, and its chemical composition also changed significantly; for example, the SO42- concentration decreased more. Therefore, in the corresponding position of the manuscript, we corrected the expression "ammonia-rich environment" and revised the relevant contents. ——————————————————————————————————————————- Page 3 line 69 "For example, photochemistry may affect the formation of NSA at high solar radiation,

and the homogeneous reaction may dominate the formation of NSA in high relative humidity" I think you mean "heterogeneous" instead of homogeneous? Response: We appreciate your comments and apologize for our unprofessional description. Our purpose is to express the complex characteristics of NSA chemical conversions. According to previous studies, both homogeneous and heterogeneous reactions have chemical conversion processes of secondary inorganic aerosols, such as photochemical reactions, aerosol liquid-phase oxidation environments, and mineral dust catalysis. Therefore, we rewrote this sentence. Now, it reads as follows: In addition, the chemical conversion of NO2, SO2 and NH3 to form NSA is still very complex, and both homogeneous and heterogeneous reactions involve the chemical conversion of secondary inorganic aerosols, such as photochemical reactions, aqueous phase oxidation environments of aerosols and catalysis of mineral dust. ——————————————————————————————————————————- Page 3 line 82 "The characteristics of higher concentrations proportion of nitrate, sulfate and ammonium in PM2.5 were also found in other polluted areas in China, such as Beijing-Tianjin-Hebei, the Yangtze River Delta, the Pearl River Delta, the Fenwei Plain, "You may not use "other" since you also refer to Beijing Tianjin-Hebei. Response: We appreciate your comments and apologize for our unprofessional description. Now, the text reads as follows: Higher concentrations of NSA in PM2.5 were also found in regions with more serious air pollution in China, such as Beijing-Tianjin-Hebei, the Yangtze River Delta, the Pearl River Delta, the Fenwei Plain, and the Chengdu-Chongqing region. ——————————————————————————————————————————- Page 5 line 123, Sect 2.1 According to the high NO concentrations, the site could be quite close to adjacent sources. This should be mentioned in the site description. Response: We appreciate your comments and apologize for our unprofessional description. As you believe, our observation station is located in the central area of the city, and the contribution of vehicle emissions may also be prominent, so we revised the description of the observation station. Now, it reads as follows: Comprehensive observations were carried out at the Chengdu comprehensive observation station of atmospheric

combined pollution (30.63°N, 104.08°E). The observation equipment was placed on the top of a building, approximately 25 m from the ground, and there was no obvious pollution source within approximately 200 m. The site is located in south section 1 of Yihuan Road, Wuhou District, Chengdu (Fig. 1), and traffic emission sources may be the main pollution emission source around the observation station. This is a typical residential, traffic and commercial mixed area that represents the characteristics of the urban atmospheric environment. ————————————————————————————

———————————————- Page 7 line 154 "Temperature (T), relative humidity (RH) and the total concentrations (i.e., gas + aerosol) of Na+, SO42-, NH3, NO3-, Cl-, 155 Ca2+, K+ and Mg2+ were input into the ISORROPIA-II thermodynamic mode" Do you have HCl, and HNO3 measured? I don't see it in your instrument list. You may need to do a back calculation to check the modelled value and see if you may retrieve these information iterative model calculations. You may need to calculate the uncertainties or bias due to these missing data in your model input. Response: We appreciate your comments and apologize for our unprofessional description. Your comment was very helpful for us in improving this research, and it also reminded us to pay attention to the key points when entering data in the ISORROPIA-II thermodynamic model. Therefore, we re-simulated and analysed the the model. In our observation experiment, HCl and HNO3 were not measured, so the data of these two species could not be obtained. Therefore, in our data input, Cl- and NO3- are entered, and NH3 is the total ammonium (NH3+NH4+) of gas and aerosol inputs. We used this model to analyse the observation and simulation data of NSA in metastable and stable state conditions in forward mode. We used NH3 model simulation data and observation data to perform linear regression fitting to verify the model run effect, and the fitting slope of linear regression is 0.96 (R2=0.98), which shows that the model effect is also good, and it can reflect the state of chemical components in aerosols. To reduce the uncertainties or bias caused by missing data, through strict data quality control during data input, we ensured the integrity and validity of each sample data, 618 samples were simulated, and the content of this model in the current manuscript

was also re-described. ———————————————————————————————
—————————————- Page 7 line 159 "The simulated data and observed data were compared and analysed. Simultaneously, the aerosol water content (AWC) and pH of aerosols were calculated. The sensitivity of the interaction between aerosol chemical components (NSA) was analysed (Ding et al., 161 2019;Fountoukis et al., 2009). Could you show a comparison between the modelled and measured gas phase NH3, HCl and HNO3? This result can be used to check the reliability and performance of thermodynamic models. Response: We appreciate your comments and apologize for our unprofessional description. Through data quality assurance and control, combined with our research purpose, the usage of thermodynamic balance in this manuscript is modified. We agree that it is necessary to analyse the reliability and performance of the output results of the model. Therefore, we performed a comparative analysis of the output data and the observation data. Considering that the observation experiment did not measure HCl and HNO3, we only analysed the observation and simulation data of NH3 and compared the observation and simulation data of NSA. In addition, in section 2.4 of the manuscript, we also give an accurate description of the model and the methods used in this study. In response to the current comment, we made the following revision in section 2.4: The simulated data and observed data were compared and analysed, and the observation data of NH3 were consistent with the input data of the model. The linear regression fitting slope of NH3 was 0.96 (R2=0.98), which showed that the run result of the model had good reliability and performance. —————————————————————————————————————————- Page 8 line 178 "the conditional probability function (CPF) was introduced the R Programming Language." Complete the sentence. Response: We appreciate your comments and apologize for our unprofessional description. Now, the text reads as follows: We used the conditional probability function (CPF) to analyse the characteristics of pollutants under the influence of wind direction (WD) and wind speed (WS). The analysis results using CPF were obtained using the R programming language, named openair. —————————————————————————————————————————- Figure

2, why both fractional contribution of both organic and inorganic decrease at high PM2.5 concentrations? What's the other compositions that are increasing? Response: We appreciate your comments and apologize for our unprofessional description. We were also initially puzzled by this problem, but by reading many research results, we also obtained a deep understanding. In our research, the proportion of organic (OC and EC) and inorganic (NSA) components in PM2.5 is analysed, which will certainly involve a large number of chemical components that have not been calculated and measured, which also reflects the complexity of the composition of PM2.5 chemical components. This variation in the chemical composition of PM2.5 has also been confirmed in studies in other regions of China. A long-term observation of OC and EC in PM2.5 from 2013 to 2018 in Beijing shows that with the accumulation of PM2.5 concentration, the concentrations of OC and EC increased, and the proportion of PM2.5 decreased (Ji et al., 2019). In a series of research reports on the evaluation of the Air Pollution Prevention and Control Action Plan (2013-2017), in the Chengdu-Chongqing region, the concentration of PM2.5 gradually decreased from 2013 to 2015, and the proportion of NSA in PM2.5 gradually increased, which also shows that when pollution is aggravated, the chemical composition of higher PM2.5 concentrations is more complex, and the unknown component will contribute to a certain quality (Wang et al., 2019). We have revised the corresponding part in the manuscript. Now, it reads as follows: This phenomenon occurs because some chemical components are included in the statistical analysis. It also reflects that the chemical components of PM2.5 have more complex characteristics when pollution is aggravated. Some studies have analysed the changes in the chemical composition of particulate matter in regions with severe pollution in China in recent years, and the results show that the concentration of particulate matter has been significantly reduced, but other components (except NSA and carbonaceous aerosol) have higher contribution characteristics at higher particle concentrations (Geng et al., 2019;Wang et al., 2019). The variation trend of OC, EC and metal elements with increasing PM2.5 concentration is similar to that of NSA (Fig. 2c), and this variation trend of OC and EC is consistent with the results of long-term

observation research carried out in Beijing (Ji et al., 2019). Reference: Geng, G., Xiao, Q., Zheng, Y., Tong, D., Zhang, Y., Zhang, X., Zhang, Q., He, K., and Liu, Y.: Impact of China's Air Pollution Prevention and Control Action Plan on PM2.5 chemical composition over eastern China, Science China Earth Sciences, 62, 1872-1884, 10.1007/s11430-018-9353-x, 2019. Ji, D., Gao, W., Maenhaut, W., He, J., Wang, Z., Li, J., Du, W., Wang, L., Sun, Y., Xin, J., Hu, B., and Wang, Y.: Impact of air pollution control measures and regional transport on carbonaceous aerosols in fine particulate matter in urban Beijing, China: insights gained from long-term measurement, Atmospheric Chemistry and Physics, 19, 8569-8590, 10.5194/acp-19-8569-2019, 2019. Wang, Y., Li, W., Gao, W., Liu, Z., Tian, S., Shen, R., Ji, D., Wang, S., Wang, L., Tang, G., Song, T., Cheng, M., Wang, G., Gong, Z., Hao, J., and Zhang, Y.: Trends in particulate matter and its chemical compositions in China from 2013-2017, Science China Earth Sciences, 62, 1857-1871, 10.1007/s11430-018-9373-1, 2019.

––––––––––––––––––––––––––––––––––––––––––––––––––––––– Page 9

line 216 "The annual average mass concentration of NSA also changed significantly, and the difference was large. The Mann-Whitney U test showed that the variation in NO3 - was nonsignificant (p > 0.05), and SO4 2- and NH4 + had obvious significance from 2015 to 2017 (p < 0.05), indicating that NO3 - had not decreased significantly, and there was an increase in 2017 compared to 2015." Here, you could further discuss the reasons why the concentration SO2 and sulfate decease more than that of NOx and nitrate. Response: We appreciate your comments and apologize for our unprofessional description. This similar problem has also been raised by Anonymous Reveiwer #1. We believe that the current greater emission reduction efforts are due to the implementation of the Air Pollution Prevention and Control Action Plan, and a series of pollution control measures have been implemented. Therefore, we have added in the manuscript the emission reduction and control measures taken by Sichuan Province in recent years to control air pollution. The concentrations of SO2 and SO42- decease more than those of NOx and NO3-, indicating that it is necessary to strengthen the air treatment for NOx emissions. We have made corresponding

revisions in the manuscript. ————————————————————————————
————————————— Page 12, line 283 "This also shows that the implementation
of air pollution reduction measures should increase the emission reduction intensity
in terms of NOx and NH3 emissions, especially the implementation of autumn and
winter air pollution prevention and control action." You were talking about high NH3
in 2017 and then taking to NOx? I am missing a link here. Also I'd like to see an
explanation about the high NH3 concentration up to 60 ppb. This is very high for a
monthly average. What's the pH under this condition? Response: We appreciate your
comments and apologize for our unprofessional description. We are here to make a
brief summary of the monthly change trend of gaseous pollution; because we have
revised a mistake in expression that caused ambiguity. We also noticed that in the
second half of 2017, there was a higher NH3 emission intensity, especially in winter. In
combination with Section 3.5, we also found that there will be a higher concentration
of pollutants at lower wind speeds (CPF analysis). There may be an emission source
in the nearby area that played a more significant role in 2017. In addition, the PSCF
analysis also has an obvious regional transport source in the northeast direction.
The contribution in this direction is significantly higher than that in 2015 and 2016.
Therefore, we believe that the higher NH3 emissions in 2017 are affected by local
emissions and regional transport. Since chemical transport models were not available
in this study to quantitatively analyse the contribution of local emissions and regional
transport, it is necessary for us to conduct in-depth research in this area in the future.
We recalculated and analysed the ISORROPIA-II thermodynamic equilibrium model
through data quality assurance and control. Because the data were not suitable for
annual change analysis, the pH value of aerosols at higher NH3 concentrations in
2017 was not analysed. ————————————————————————————
————————————— Page 13 line 293 "Sulfate has a significant downward trend in all
seasons from 2015 to 2017, especially in winter. This downward trend was due to
implementation of the Air Pollution 294 Prevention and Control Action Plan" Such
discussion should be put in Sect 3.1. Response: We appreciate your comments and

apologize for our unprofessional description. We have adjusted the position of this discussion and revised the manuscript. ————————————————————————————— ——————————————————————————- Page 15 line 325 "As shown in Fig. S4, from 9:00 to 11:00 a.m., the concentrations of SO2, NOx, NH3, CO and other gases increased significantly, indicating that the primary emission of pollutants was relatively strong. At this time, higher RH (Fig S5) also provides favourable conditions for the formation of secondary aerosols and promotes the accumulation of NSA" But the RH in Fig. S5 is decreasing in contrast to an increase in aerosol concentrations? Response: We appreciate your comments and apologize for our unprofessional description. As you have noticed, RH is indeed decreasing from 9:00 to 11:00. In our analysis, despite the decrease, RH was still relatively high (approximately 65%). Therefore, we have revised and supplemented this description. Now, it reads as follows: As shown in Fig. S7, from 9:00 to 11:00 a.m., the concentrations of SO2, NOx, NH3 and CO increased significantly, indicating that the primary emission of pollutants was relatively strong. At this time, although RH is in a declining stage, it still has a relatively high atmospheric humidity (approximately 65%), and O3 and NO2/NO also occasionally show an increasing trend, indicating that the atmospheric oxidizability has also increased (Figs. S7 and S8). This situation also provides favourable conditions for the formation of secondary aerosols and promotes the accumulation of NSA. ——————————————————————————————————————————————————- Sect. 3.4.1 In general, it is true that the emissions of multi-pollutant may come from the same kinds of sources. But you cannot draw such a conclusion based on correlation studies. Because the variation of most pollutants, especially those of long-lifetime, is strongly influenced by the boundary layer developments, and may show a similar diurnal variation in spite of different origins (sources). Response: We appreciate your comments and apologize for our unprofessional description. We agree with you. During the ACP discussion stage of the manuscript, our research team also carefully reviewed this text and found that the current analysis is not appropriate and may affect the integrity of the manuscript. Therefore, we decided to delete this section.

20 Line 430 "Figure 7 shows the variation characteristics of NSA chemical conversions and meteorological conditions with increasing RH. SOR and NOR increased with increasing RH, suggesting that SO2 and NO2 were more likely to produce sulfate and nitrate under higher RH conditions. In Fig. 7, how did you do the calculation, classifying the data according to RH or you keep all input the same but change RH only? In the former case, the apparent correlation with RH may not represent the real causation as chemical compositions and other parameters may change also change. Response: We appreciate your comments and apologize for our unprofessional description. We have revised this text. In response to your concerns, we classified and statistically analysed the variation characteristics of NOR and SOR under different RH conditions according to RH observation data. Regarding the latter comment you raised, we also very much agree with you. The correlation is not enough to explain the relationship between chemical components and other parameters. Therefore, we combined the phase state of the chemical components analysed by the ISORROPIA-II thermodynamic equilibrium model to supplement the analysis.

—————————————————————————————————————————————————- Sect

3.5.2 I understand that the authors adopted this approach based on a published study. This approach, however, is subject to several problems, e.g., neglecting the dilution of pollution in the course of transport which may overestimate the contribution of distant sources, or the endpoint is not necessary at the ground level, or why 24 hour (aerosols have a longer lifetime) etc. If you still want to keep this part, please explicitly include these caveats in the text to avoid misinterpretation of this result. Response: We appreciate your comments and apologize for our unprofessional description. As you pointed out, aerosols do influence of emissions, diffusion, chemical conversions and deposition in the process of regional transport. In our study, PSCF is a kind of conditional probability function relationship. Using Meteorological data from the National Oceanic and Atmospheric Administration (NOAA) to analyse the potential sources of pollution is helpful for explaining the importance of regional joint prevention

and control measures for air pollution control. We choose a 24-hour simulation time, mainly considering the following factors. The aerosol spatial distribution characteristics show that there is a high concentration of pollutants in the Sichuan Basin in Southwest China (Gui et al., 2019), and due to the unique topography of the Sichuan Basin, air pollution is obviously affected by the internal emission (Qiao et al., 2019). We agree that the endpoints of the backward trajectory are not on the ground, in fact, as you think. The results of PSCF reflect the potential source of pollution in a plane, rather than the three-dimensional spatial structure characteristics, and the endpoints of the backward trajectory are reflected in the design plane grid, which also better reflects the high-value regional distribution features of PSCF. Therefore, the PSCF reflects the two-dimensional planar position distribution characteristics of potential sources, not the three-dimensional characteristics that reflect the transmission of pollution. In addition, the aerosol lifetimes of SO2 (approximately 9.6 d) and NOx (approximately 1 d) are also very different (Guo et al., 2014), and the research also shows that NH3 is significantly contributed by local source emissions (Walker et al., 2004). Therefore, we comprehensively considered selecting a 24-hour backward trajectory to carry out PSCF simulation in the Chengdu region. The corresponding supplementary notes have been revised in the manuscript. Reference: Gui, K., Che, H., Wang, Y., Wang, H., Zhang, L., Zhao, H., Zheng, Y., Sun, T., and Zhang, X.: Satellite-derived PM2.5 concentration trends over Eastern China from 1998 to 2016: Relationships to emissions and meteorological parameters, Environmental pollution, 247, 1125-1133, 10.1016/j.envpol.2019.01.056, 2019. Qiao, X., Guo, H., Tang, Y., Wang, P., Deng, W., Zhao, X., Hu, J., Ying, Q., and Zhang, H.: Local and regional contributions to fine particulate matter in the 18 cities of Sichuan Basin, southwestern China, Atmospheric Chemistry and Physics, 19, 5791-5803, 10.5194/acp-19-5791-2019, 2019. Guo, S., Hu, M., Zamora, M. L., Peng, J., Shang, D., Zheng, J., Du, Z., Wu, Z., Shao, M., Zeng, L., Molina, M. J., and Zhang, R.: Elucidating severe urban haze formation in China, Proceedings of the National Academy of Sciences of the United States of America, 111, 17373-17378, 10.1073/pnas.1419604111, 2014. Walker, J. T., Whitall,

D. R., Robarge, W., and Paerl, H. W.: Ambient ammonia and ammonium aerosol across a region of variable ammonia emission density, Atmospheric Environment, 38, 1235-1246, 10.1016/j.atmosenv.2003.11.027, 2004. ————————————————
————————————————————————————————-

Please also note the supplement to this comment:
https://www.atmos-chem-phys-discuss.net/acp-2019-1142/acp-2019-1142-AC2-supplement.pdf

————————————————————————

[Figure]

[Figure]

**Fig. 1.** Fig. S1. PM10, PM2.5 and PM1 data quality assurance and control.

[Figure]

**Fig. 2.** Fig. S2. NO3-, SO42-, NH4+, OC (organic carbon) and EC (element carbon) data quality assurance and control.

[Figure]

**Fig. 3.** Fig. S3. NOx, SO2, NO2, NO, CO and NH3 data quality assurance and control.

[Figure]

**Fig. 4.** Fig. S4. Relative humidity (RH), temperature (T), atmospheric pressure (P), wind speed
(WS) and wind direction (WD) data quality assurance and control.

---

## Author Response (AR1)

**Response to Reviewer Comments**

Dear Reviewer,

We would like to thank you for your great effort and detailed work on this manuscript.

We have revised the manuscript and responded to each of the comments from the reviewers. In our response, your questions are shown in *italics*, and the responses are shown in standard text. For the ACP discussion, our research team also performed further analysis of the research results and made minor modifications to this manuscript.

We appreciate your help and time.

Sincerely yours,

Xingang Liu and coauthors.

School of Environment

Beijing Normal University

100875 Beijing China

E-mail: liuxingang@bnu.edu.cn; lxgstar@126.com

Tel: +86-13810193569
* * *
**Title: Elucidating the pollution characteristics of nitrate, sulfate and ammonium in PM$_{2.5}$ in Chengdu, southwest China, based on three-year measurements**
* * *
**Response to Anonymous Referee#1**

*This study provided a good data basis for explaining the pollution characteristics of secondary inorganic aerosols in PM2.5 through a long-term atmospheric observation experiment. The formation mechanism and role of secondary inorganic aerosols during the formation of haze are still a research hot issue. The author not only explained the pollution characteristics of nitrate, sulfate and ammonium in Chengdu, but also analyzed its formation mechanism through observation data and ISORROPIA-II thermodynamic model simulation. Finally, the author also analyzed the distribution characteristics of pollution emissions and potential sources in Chengdu. At present, China is strengthening the control and treatment of air pollution, a long-term atmospheric observation experiment has a high research value for the analysis of the formation of air pollution and the implementation of haze abatement measures. However, there are some writing, grammatical and technical errors in the paper, and it is suggested that the author carefully revise and organize the presentation of the paper.*

**Response:**

We appreciate your comments and have revised the full text; we also regret each error and have discussed and corrected them.
* * *
*Line. 53-54, PM2.5 interpretation is inaccurate. "PM2.5(aerodynamic diameter less than 2.5 $\mu$m)"*

**Response:**

We appreciate your comments and apologize for our unprofessional description. Now, it reads as follows:

particles with aerodynamic equivalent diameter $\leq$ 2.5 μm in ambient air (PM$_{2.5}$, also known as fine particles)
* * *
*Line. 58-61, "NSA" is the abbreviation of nitrate, sulfate and ammonium in the paper? please rewrite this sentence.*

**Response:**

We appreciate your comments and apologize for our unprofessional description. Now, it reads as follows:

Nitrate, sulfate, ammonium, organic matter and elemental carbon are the main components of PM$_{2.5}$, among which nitrate, sulfate, and ammonium (NSA) are the main secondary inorganic aerosols in PM$_{2.5}$.
* * *
*Line. 77-78, suggest reinterpreting this sentence, how to understand the "regional transport"*

**Response:**

We appreciate your comments and apologize for our unprofessional description. Now, it reads as follows:

In addition to the air pollution caused by the local emission of pollutants, the regional transportation of pollutants from its surrounding cities also has an important impact on the urban air quality.
* * *
*Line. 82-83, if the author defines an abbreviation at the beginning of the paper, it is recommended to use the abbreviation below. Please use "NSA" abbreviations instead of nitrates, sulfates and ammonium.*

**Response:**

We appreciate your comments and have revised the manuscript. Now, it reads as follows:

Higher concentrations of NSA in PM$_{2.5}$ were also found in regions with more serious air pollution in China, such as Beijing-Tianjin-Hebei, the Yangtze River Delta, the Pearl River Delta, the Fenwei Plain, and the Chengdu-Chongqing region.
* * *
*Line. 87, please correct this writing, "(2013-207)"*

**Response:**

We appreciate your comments and have revised the text. Now, it reads as follows:

In response to this situation, the Chinese government issued an Air Pollution Prevention and Control Action Plan (2013-2017) in 2013 to reduce pollutant emissions and improve air quality.
* * *
*Line. 106-109, "high time resolution", what's the meaning of this?*

**Response:**

We appreciate your comments and apologize for our unprofessional description. We mean that the time interval for gathering observation data is relatively small, at 1 hour. Compared with the daily average data of manual operation sampling, the time resolution is higher; we have revised and polished this sentence. Now, it reads as follows:

However, these analyses may be affected by the experimental equipment, observation stations and other conditions, and the time span of these atmospheric observations usually includes several pollution processes or lasts for weeks or months. Thus, it is difficult to analyse the long-term variations in characteristics of air pollution through comprehensive observation. In particular, there are few high-time-resolution (1 hour) observation experiments carried out with online automatic observation systems.
* * *
*Line. 141, in Table 1, parameters not covered in this paper can be removed, such as PM1 and H2S*

**Response:**

We appreciate your comments and have revised it. Now, it reads as follows:

Table 1. The experimental instruments used in this study

| Instruments | Parameters | Manufacturer/Country |
|---|---|---|
| URG-9000 | $NO_3^-/SO_4^{2-}/NH_4^+/Na^+/Mg^{2+}/Ca^{2+}/Cl^-/K^+$ | Thermo Fisher Scientific/USA |
| SHARP 5030 | $PM_{2.5}$ | Thermo Fisher Scientific/USA |
| RT-4 | OC/EC | Sunset Laboratory/USA |
| Xact-625 | Metal elements | Cooper Environmental Services /USA |
| 17i/450i/48i/49i | $NOx/NO_2/NO/NH_3/SO_2/CO/O_3$ | Thermo Fisher Scientific/USA |
| WXT520 | Meteorological parameters | VAISALA/Germany |

OC: organic carbon; EC: element carbon
* * *
*Line. 185-186, percentile (e.g. 0-25, 25-50, 50-75 and 75-100), please confirm it is consistent with the title of Fig. 13-15 (0-25%, 25-50%, 50-75%, and 75-100%.) in Supplementary materials.*

**Response:**

We appreciate your comments and have revised the text in the Supplementary Materials. We have made the writing consistent throughout the manuscript and adopted percentiles (e.g., 0-25, 25-50, 50-75 and 75-100), and the "%" in the figure title was deleted.
* * *
*Line. 185-186, it is suggested that the author briefly describe what measures should be taken.*

**Response:**

We appreciate your comments. There is no description of "measures" in line 185, and we presume that the reviewer refers to the description in lines 228-230 about the measures taken in the Air Pollution Prevention and Control Action Plan. We also describe this part in detail. Now, it reads as follows:

From 2015 to 2017, the measures taken by Sichuan Province in the coordinated reduction of multiple pollutants have been continuously strengthened, and the scope of management and control has been continuously expanded, for example, in the improvement of desulfurization, denitrification and dust removal technologies in key industries, from accelerated improvement in 2015 to deeper improvement in 2017. The

process of eliminating small coal-fired boilers began in 2015 and was completed in 2017, when the ultra-low-emission coal-fired power plant transformation was promoted. In terms of vehicle emission control, we accelerated the elimination of "yellow label" vehicles (general term for gasoline vehicles with emission levels lower than the national I emission standard and diesel vehicles with emission levels lower than the national III emission standard when new vehicles are finalized) and "old vehicles" (the emission level does not meet the national stage IV emission standard) in 2015 and basically completed the elimination of yellow standard vehicles in 2017. The quality supervision of oil products has also been improved, and non-road mobile machinery pollution control requirements were proposed in the 2017 plan (ThePeople'sGovernmentofSichuanProvince, 2015, 2016, 2017).

**References:**

The People's Government of Sichuan Province. Detailed rules for the implementation of the action plan for the prevention and control of air pollution in Sichuan Province 2015 annual implementation plan. Website : http://www.sc.gov.cn/10462/10883/11066/2015/4/22/10333390.shtml,last access: June 17 2020.

The People's Government of Sichuan Province. Detailed rules for the implementation of the action plan for the prevention and control of air pollution in Sichuan Province 2016 annual implementation plan. Website : http://www.sc.gov.cn/zcwj/xxgk/NewT.aspx?i=20160401095908-612769-00-000 , last access: June 17 2020.

The People's Government of Sichuan Province. Detailed rules for the implementation of the action plan for the prevention and control of air pollution in Sichuan Province 2017 annual implementation plan. Website : http://www.sc.gov.cn/zcwj/xxgk/NewT.aspx?i=20170527091543-450025-00-000 , last access: June 17 2020.
* * *
*Line. 263, "These variations have similar trends due to meteorological factors", what do you mean?*

**Response:**

We appreciate your comments and apologize for our unprofessional description. What we want to say is that the meteorological conditions also have obvious monthly variation characteristics, which may have some influence on the variation characteristics of NSA. Now, the text reads as follows:

The monthly variation characteristics of NSA from 2015 to 2017 are shown in Fig. 3. At the beginning and end of each year, the pollutant concentration is relatively high and relatively low in the middle of each year. The meteorological conditions also have obvious monthly variation characteristics (Fig. S5 a and b); from April to August, they have higher WS and lower RH, which is not only conducive to the dilution and diffusion of pollutants but also reduces the chemical conversions of pollutants by aqueous phase and influences the formation of secondary inorganic aerosols.
* * *
*Line. 306, the legend in Fig. 3 is suggested to be modified, with one reserved, and also pay attention to modify other pictures, such as Fig.S2 and S8.*

**Response:**

We appreciate your comments and apologize for our unprofessional description. We reviewed similar problems in the other figures and have corrected them.
* * *
*Line. 309, replace "daily changes" with "diurnal variation".*

**Response:**

We appreciate your comments and have revised the text.
* * *
*Line. 400, please explain what "r" in Fig. 5 means?*

**Response:**

We appreciate your comments and apologize for our unprofessional description. This text has been revised in the manuscript, and this error will not appear in the new manuscript.
* * *
*Line. 452, in Fig.7h, "SO42- gas-particle phase partitioning"? inconsistent with the*

*NH4+ in the picture.*

**Response:**

We appreciate your comments and apologize for our unprofessional description. The analysis content has been modified in this section, and the comments made by you in the manuscript have also been resolved.
* * *
*Line. 470, please explain what "k" in Fig. 8 means?*

**Response:**

We appreciate your comments and apologize for our unprofessional description. Now, the text reads as follows:

Fig. 8. Molar ratio analysis of NSA (nitrate, sulfate and ammonium). (a) Interannual variation in the molar ratio of $SO_4^{2-}$ and $NH_4^+$. (b) Interannual variation in the molar ratio of $NO_3^-$ and $NH_4^+$. (c) Seasonal variation in the molar ratio of $SO_4^{2-}$ and $NH_4^+$. (d) Seasonal variation in the molar ratio of $NO_3^-$ and $NH_4^+$. k: Fitting slope of linear regression.
* * *
*In Section 3.5.2, authors are advised to supplement PSCF analysis of NO2 and NO.*

**Response:**

We appreciate your comments and have revised the text.
* * *
*Line. 470, modify the Fig.9, remove the repeat "PSCF" in the picture*

**Response:**

We appreciate your comments and apologize for our unprofessional description. This issue is also associated with the previous issue, and we have solved the problem.
* * *
*Line. 521-525, the description is too simple, please rewrite the research results.*

**Response:**

We appreciate your comments and apologize for our unprofessional description. Now, the text reads as follows:

The results of using the ISORROPIA-II thermodynamic equilibrium model to simulate

$NO_3^-$, $SO_4^{2-}$ and $TNH_3$ emission reduction control effects of 5%, 10%, 15% and 20%, respectively, are shown in Table S3, showing that controlling the concentration of $NO_3^-$ and $SO_4^{2-}$ is also helpful to reduce the concentration of $NH_4^+$ and indicating that controlling its precursor NOx and $SO_2$ is of great significance to reduce the secondary inorganic aerosol in $PM_{2.5}$ (the detailed results are described in the supplementary materials).

*supplementary materials*

Through observation data quality control, we screened 618 sample input ISORROPIA-II thermodynamic equilibrium models to ensure the integrity of the samples and the effectiveness of the data. The control variable method was used to explore the impact of a concentration reduction for other species. For example, to explore the impact of $NO_3^-$ concentration reduction for $SO_4^{2-}$ and $NH_4^+$, the $NO_3^-$ data were calculated based on the 5, 10, 15 and 20% emission reduction ratio, and other parameters were input into the model using the observation data to explore the relative variable of $SO_4^{2-}$ and $NH_4^+$ concentration. The simulation results are shown in Table S3. When only $NO_3^-$ and $SO_4^{2-}$ were reduced, $NH_4^+$ was significantly reduced, but the changes in $SO_4^{2-}$ and $NO_3^-$ were not obvious, and the relative variables of approximately 12% and 7% may be mainly affected by the change in phase state. When only $TNH_3$ was controlled, the relative variable of $SO_4^{2-}$ was not obvious, and the concentrations of $NO_3^-$ and $NH_4^+$ decreased, but the relative variable was not large. NSA has a good reduction effect under synergistic emission reduction control. The results show that reducing the amount of $NO_3^-$ and $SO_4^{2-}$ can not only reduce their concentrations but also help to reduce the concentration of $NH_4^+$. It also suggests that controlling the gaseous precursors NOx and $SO_2$ is of great significance to reduce the amount of secondary inorganic aerosol in $PM_{2.5}$. Studies in Mexico City have also shown that reducing total sulfates and total nitrates rather than total ammonium helps reduce $PM_{2.5}$ concentrations in an ammonium-rich environment (Fountoukis et al., 2009).

Table S3. Simulation of $NO_3^-$, $SO_4^{2-}$ and $TNH_3$ emission reduction control effect (%) and its influence on pH based on the ISORROPIA-II thermodynamic model.

| Reduction | Only NO$_3^-$ Reduction | | | | Only SO$_4^{2-}$ Reduction | | | |
|---|---|---|---|---|---|---|---|---|
| | NO$_3^-$ | SO$_4^{2-*}$ | NH$_4^+$ | pH$^*$ | NO$_3^-$ | SO$_4^{2-*}$ | NH$_4^+$ | pH |
| 5% | 11.92 | 12.25914 | 8.33 | 4.0495 | 7.19 | 17.1088 | 9.77 | 4.08 |
| 10% | 16.58 | 12.25911 | 11.13 | 4.0519 | 7.09 | 21.9593 | 13.65 | 4.13 |
| 15% | 21.23 | 12.25909 | 13.91 | 4.0546 | 7.00 | 26.8093 | 17.50 | 4.19 |
| 20% | 25.88 | 12.25906 | 16.69 | 4.0547 | 6.91 | 31.6596 | 21.58 | 4.25 |
| | Only TNH$_3$ Reduction | | | | Synergistic$^{**}$ | | | |
| | NO$_3^-$ | SO$_4^{2-*}$ | NH$_4^+$ | pH | NO$_3^-$ | SO$_4^{2-*}$ | NH$_4^+$ | pH |
| 5% | 7.51 | 12.25938 | 5.85 | 4.02 | 12.08 | 17.1090 | 12.86 | 4.07 |
| 10% | 7.79 | 12.25965 | 6.20 | 3.99 | 16.85 | 21.9596 | 19.80 | 4.09 |
| 15% | 8.10 | 12.25998 | 6.59 | 3.95 | 21.64 | 26.8097 | 26.17 | 4.11 |
| 20% | 8.45 | 12.26040 | 7.03 | 3.91 | 26.37 | 31.6601 | 33.29 | 4.16 |

Notes: NO$_3^-$, SO$_4^{2-}$ and TNH$_3$ are the concentration variables relative to the observation data;

pH is the average; TNH$_3$: NH$_3$ + NH$_4^+$;

*: In order to display the data difference, the number of digits after the decimal point was increased

**: NO$_3^-$, SO$_4^{2-}$ and TNH$_3$ decreased in the same proportion

**Reference:**

Fountoukis, C., Nenes, A., Sullivan, A., Weber, R., Van Reken, T., Fischer, M., Matias, E., Moya, M., Farmer, D., and Cohen, R. C.: Thermodynamic characterization of Mexico City aerosol during MILAGRO 2006, Atmospheric Chemistry and Physics, 9, 2141-2156, 10.5194/acp-9-2141-2009, 2009.
* * *
*Authors are requested to write rules uniformly. It is not recommended to use abbreviations in the title of the figures, such as NSA, AWC, SOR, NOR PSCF. In addition, there are "(a)", "(b)" and "(c)" in the picture, please explain what it means in the title.*

**Response:**

We appreciate your comments and apologize for our unprofessional description. We have checked and corrected the abbreviations for consistency throughout the

manuscript.
* * *
*In Fig. 5 and Fig. S4, please confirm the carbon monoxide (CO) unit, "ppb" and "ppm"?*

**Response:**

We appreciate your comments and apologize for our unprofessional description. The unit of carbon monoxide (CO) is ppm, and we have revised it.
* * *
*In Supplementary materials, Line. 90 and 96, pay attention to writing, it shouldn't be "2107"*

**Response:**

We appreciate your comments and have revised it.
* * *
**Response to Reviewer Comments**

Dear Reviewer,

We would like to thank you for your great effort and detailed work on this manuscript.

We have revised the manuscript and responded to each of the comments from the reviewers. In our response, your questions are shown in ***italics***, and the responses are shown in standard text. For the ACP discussion, our research team also performed further analysis of the research results and made minor modifications to this manuscript.

We appreciate your help and time.

Sincerely yours,

Xingang Liu and coauthors.

School of Environment

Beijing Normal University

100875 Beijing China

E-mail: liuxingang@bnu.edu.cn; lxgstar@126.com

Tel: +86-13810193569
* * *
**Title: Elucidating the pollution characteristics of nitrate, sulfate and ammonium in PM$_{2.5}$ in Chengdu, southwest China, based on three-year measurements**
* * *
**Response to Anonymous Referee#2**

*This paper presented an overview of air pollution in Chengdu, southwest China based on study, a three-year observations of gas and particulate pollutants. Probably due to the special topography, the data from this site shows special characteristics of pollutants, different from most other polluted regions in China, e.g., North China plain and Pearl River Delta etc. Thermodynamic models and trajectory analysis have also been applied to analyze aerosol pH, partitioning of inorganic semi-volatile species and the contribution of potential source regions. Overall, I think that the datasets are very interesting and valuable, and authors have tried to give a comprehensive overview and analysis of the mechanisms beyond. However, there is still much room for improvement. I would suggest the authors to carefully consider my comments/suggestions in their revision before its final publication in ACP.*

**Response:**

We appreciate your comments and have revised the manuscript accordingly. We firmly believe that your guidance is of great significance for improving our research.
* * *
**Major concern:**

*1. Meteorological parameters and PM compositions show distinct diurnal variations compared to other regions such as NCP, PRD and YRD. Though chemistry is in this game, I guess that the special topography may play a dominant role in these features, which is missing from this submission. I'd suggest the authors to add these discussions.*

**Response:**

We appreciate your comments and have revised the text. We also noticed that during the data analysis, the pollutant concentration in the daytime was higher, which was obviously different from that in NCP, PRD and YRD. We also collated the relevant research results in NCP, PRD and YRD and found that they were consistent with the

questions you raised. Therefore, according to your guesses and prompts, we conducted an in-depth analysis. The unique topographic structural features did have an important impact on the diurnal variation of meteorological factors in the Sichuan Basin, which may also be an important factor that causes the daily changes in air pollutants with unique characteristics. We also supplement and discuss the corresponding parts of the manuscript. Now, the text reads as follows:

In previous studies in Beijing-Tianjin-Hebei and the Pearl River Delta, the concentration of pollutants was affected by meteorological factors, and it was usually lower in the daytime than at night. In the Yangtze River Delta, the peak usually occurs in the morning, but in our study, the concentration was higher in the daytime than at night (Peng et al., 2011;Wang et al., 2018;Guo et al., 2017b). In addition to the diurnal variations in WS and atmospheric humidity, some studies have shown that due to the unique topographical structure of the Sichuan Basin, the atmospheric circulation between the Qinghai-Tibet Plateau, Yunnan-Guizhou Plateau and Sichuan Basin and the meteorological conditions of the Chengdu region are affected, such as the characteristics of air mass transport and typical "night rain" (more precipitation at night than in the day) under the influence of atmospheric circulation (Zhang et al., 2019b;Zhang et al., 2019a).

**Response:**

We appreciate your comments and apologize for our unprofessional description. We agree with you very much because the instruments involved in this study are online monitoring equipment and have high time resolution (1 hour), so data quality assurance and control are the keys to determining the accuracy and scientific nature of this study. Therefore, we added this information to the supplementary materials. Now, it reads as follows:

Data quality control and assurance are important components of atmospheric comprehensive observation experiments. In addition to regular inspection and correction of the equipment through professional operation and maintenance to ensure the accuracy of experimental data, the quality control and processing of monitoring data, such as excluding outliers and data beyond the detection limit, are also an important. As shown in Figs. S1-4, the time sequence of monitoring data and the red part in the figure indicate that the data are missing and that the overall data integrity is good. The missing rate of $PM_{2.5}$ data in Fig. S1 is 6.8%. The missing rates of $NO_3^-$, $SO_4^{2-}$, $NH_4^+$,

OC and EC data in Fig. S2 are 18.3, 17.1, 20.9, 15.2 and 19.6%, respectively. In Fig. S3, the gaseous pollution of NO data is missing 18.2%, NH$_3$ is missing 11.3%, and other gases are missing 9%. The quality of meteorological data is good (Fig. S4), and the overall missing rate is 3.1% or less. On the whole, the observation data are good and do not affect the continuity of the data as a whole. The Cl$^-$, Na$^+$, K$^+$, Mg$^{2+}$ and Ca$^{2+}$ data are significantly missing, and this study only involved in analysis of the ISORROPIA-II thermodynamic equilibrium model. To ensure that each sample data point can be input into the model completely, 618 sample input models are selected according to the data quality control to eliminate the impact of missing ion data to ensure that the model analysis results are effective.

[Figure]

Fig. S1. PM$_{2.5}$ data quality assurance and control.

[Figure]

Fig. S2. NO$_3^-$, SO$_4^{2-}$, NH$_4^+$, OC (organic carbon) and EC (element carbon) data quality assurance and control.

[Figure]

Fig. S3. NOx, SO$_2$, NO$_2$, NO, CO and NH$_3$ data quality assurance and control.

[Figure]

Fig. S4. Relative humidity (RH), temperature (T), wind speed (WS) and wind direction

(WD) data quality assurance and control.
* * *
*3. To avoid jump between text and SI, the authors could consider moving some SI parts into the main text. For example, gas phase NH3 that frequently used and discussed is missing from the main text.*

**Response:**

We appreciate your comments and apologize for our unprofessional description. We have adjusted the positions of the figures according to the structure of the whole manuscript.
* * *
**Minor comments:**

*Abstract: Line 21 "a long-term observational experiment was conducted from January 1, 2015 to December 31, 2017" Three years measurements are longer than a campaign-based experiment but I won't call it "long-term".*

**Response:**

We appreciate your comments and apologize for our unprofessional description. We agree with you that the expression "long-term" observation instead of "three-year" observation is not accurate enough, so we use "three-year" observation.
* * *
*Line 27 "Seasonal and diurnal variations have obvious characteristics, winter still has a high NSA concentration and emission intensity, and the concentration during the day was higher than that at night. "This is unusual; is it because of the valley topography*

**Response:**

We appreciate your comments and apologize for our unprofessional description. As you commented in "Major concern: 1", we have performed a comparative study of other regions of China, Beijing-Tianjin-Hebei, the Yangtze River Delta and the Pearl River Delta, and found that, as you believe, the diurnal variation of air pollutants in Sichuan Basin is indeed different from that in other regions. Based on related research results, the unique topographical structure of Sichuan Basin will indeed affect the meteorological conditions of Sichuan Basin by affecting the atmospheric circulation

(interaction of the Qinghai Tibet Plateau, Yunnan Guizhou Plateau and Sichuan Basin). Therefore, the daily variation characteristics of pollutants may be significantly affected by meteorological conditions. In response to "Major concern: 1", we have made corresponding modifications in the manuscript.
* * *
*Line 34 "The ammonia-rich environment became increasingly obvious in the atmosphere of Chengdu" It is not clear what you want to say. Do you mean that you see an increase in NH3 concentration or partition rate?*

**Response:**

We appreciate your comments and apologize for our unprofessional description. We have reconsidered our research purpose and found that the expression of the "ammonia-rich" environment is not accurate. Therefore, we revised the expression based on our current research results. In section 3.4.3 of the manuscript, the ratio of $NH_4^+$ to $NO_3^-$ and $SO_4^{2-}$ was analysed and found to increase from 2015 to 2017. The results may be attributed to the fact that the concentration of $PM_{2.5}$ gradually decreased under the implementation of relevant air pollution control measures, and its chemical composition also changed significantly; for example, the $SO_4^{2-}$ concentration decreased more. Therefore, in the corresponding position of the manuscript, we corrected the expression "ammonia-rich environment" and revised the relevant contents.
* * *
*Page 3 line 69 "For example, photochemistry may affect the formation of NSA at high solar radiation, and the homogeneous reaction may dominate the formation of NSA in high relative humidity" I think you mean "heterogeneous" instead of homogeneous?*

**Response:**

We appreciate your comments and apologize for our unprofessional description. Our purpose is to express the complex characteristics of NSA chemical conversions. According to previous studies, both homogeneous and heterogeneous reactions have chemical conversion processes of secondary inorganic aerosols, such as photochemical reactions, aerosol liquid-phase oxidation environments, and mineral dust catalysis. Therefore, we rewrote this sentence. Now, it reads as follows:

In addition, the chemical conversion of $NO_2$, $SO_2$ and $NH_3$ to form NSA is still very complex, and both homogeneous and heterogeneous reactions involve the chemical conversion of secondary inorganic aerosols, such as photochemical reactions, aqueous phase oxidation environments of aerosols and catalysis of mineral dust.
* * *
*Page 3 line 82 "The characteristics of higher concentrations proportion of nitrate, sulfate and ammonium in PM2.5 were also found in other polluted areas in China, such as Beijing-Tianjin-Hebei, the Yangtze River Delta, the Pearl River Delta, the Fenwei Plain, "You may not use "other" since you also refer to Beijing Tianjin-Hebei.*

**Response:**

We appreciate your comments and apologize for our unprofessional description. Now, the text reads as follows:

Higher concentrations of NSA in $PM_{2.5}$ were also found in regions with more serious air pollution in China, such as Beijing-Tianjin-Hebei, the Yangtze River Delta, the Pearl River Delta, the Fenwei Plain, and the Chengdu-Chongqing region.
* * *
*Page 5 line 123, Sect 2.1 According to the high NO concentrations, the site could be quite close to adjacent sources. This should be mentioned in the site description.*

**Response:**

We appreciate your comments and apologize for our unprofessional description. As you believe, our observation station is located in the central area of the city, and the contribution of vehicle emissions may also be prominent, so we revised the description of the observation station. Now, it reads as follows:

Comprehensive observations were carried out at the Chengdu comprehensive observation station of atmospheric combined pollution (30.63°N, 104.08°E). The observation equipment was placed on the top of a building, approximately 25 m from the ground, and there was no obvious pollution source within approximately 200 m. The site is located in south section 1 of Yihuan Road, Wuhou District, Chengdu (Fig. 1), and traffic emission sources may be the main pollution emission source around the observation station. This is a typical residential, traffic and commercial mixed area that

represents the characteristics of the urban atmospheric environment.
* * *
*Page 7 line 154 "Temperature (T), relative humidity (RH) and the total concentrations (i.e., gas + aerosol) of Na+, SO42-, NH3, NO3-, Cl-, 155 Ca2+, K+ and Mg2+ were input into the ISORROPIA-II thermodynamic mode" Do you have HCl, and HNO3 measured? I don't see it in your instrument list. You may need to do a back calculation to check the modelled value and see if you may retrieve these information iterative model calculations. You may need to calculate the uncertainties or bias due to these missing data in your model input.*

**Response:**

We appreciate your comments and apologize for our unprofessional description. Your comment was very helpful for us in improving this research, and it also reminded us to pay attention to the key points when entering data in the ISORROPIA-II thermodynamic model. Therefore, we re-simulated and analysed the the model. In our observation experiment, HCl and $HNO_3$ were not measured, so the data of these two species could not be obtained. Therefore, in our data input, $Cl^-$ and $NO_3^-$ are entered, and $NH_3$ is the total ammonium ($NH_3+NH_4^+$) of gas and aerosol inputs. We used this model to analyse the observation and simulation data of NSA in metastable and stable state conditions in forward mode. We used $NH_3$ model simulation data and observation data to perform linear regression fitting to verify the model run effect, and the fitting slope of linear regression is 0.96 ($R^2=0.98$), which shows that the model effect is also good, and it can reflect the state of chemical components in aerosols. To reduce the uncertainties or bias caused by missing data, through strict data quality control during data input, we ensured the integrity and validity of each sample data, 618 samples were simulated, and the content of this model in the current manuscript was also re-described.
* * *
*Page 7 line 159 "The simulated data and observed data were compared and analysed. Simultaneously, the aerosol water content (AWC) and pH of aerosols were calculated. The sensitivity of the interaction between aerosol chemical components (NSA) was analysed (Ding et al., 161 2019;Fountoukis et al., 2009). Could you show a comparison*

*between the modelled and measured gas phase NH3, HCl and HNO3? This result can be used to check the reliability and performance of thermodynamic models.*

**Response:**

We appreciate your comments and apologize for our unprofessional description. Through data quality assurance and control, combined with our research purpose, the usage of thermodynamic balance in this manuscript is modified. We agree that it is necessary to analyse the reliability and performance of the output results of the model. Therefore, we performed a comparative analysis of the output data and the observation data. Considering that the observation experiment did not measure HCl and $HNO_3$, we only analysed the observation and simulation data of $NH_3$ and compared the observation and simulation data of NSA.  In addition, in section 2.4 of the manuscript, we also give an accurate description of the model and the methods used in this study. In response to the current comment, we made the following revision in section 2.4:

The simulated data and observed data were compared and analysed, and the observation data of $NH_3$ were consistent with the input data of the model. The linear regression fitting slope of $NH_3$ was 0.96 ($R^2$=0.98), which showed that the run result of the model had good reliability and performance.
* * *
*Page 8 line 178 "the conditional probability function (CPF) was introduced the R Programming Language." Complete the sentence.*

**Response:**

We appreciate your comments and apologize for our unprofessional description. Now, the text reads as follows:

We used the conditional probability function (CPF) to analyse the characteristics of pollutants under the influence of wind direction (WD) and wind speed (WS). The analysis results using CPF were obtained using the R programming language, named openair.
* * *
*Figure 2, why both fractional contribution of both organic and inorganic decrease at high PM2.5 concentrations? What's the other compositions that are increasing?*

**Response:**

We appreciate your comments and apologize for our unprofessional description. We were also initially puzzled by this problem, but by reading many research results, we also obtained a deep understanding. In our research, the proportion of organic (OC and EC) and inorganic (NSA) components in $PM_{2.5}$ is analysed, which will certainly involve a large number of chemical components that have not been calculated and measured, which also reflects the complexity of the composition of $PM_{2.5}$ chemical components. This variation in the chemical composition of $PM_{2.5}$ has also been confirmed in studies in other regions of China. A long-term observation of OC and EC in $PM_{2.5}$ from 2013 to 2018 in Beijing shows that with the accumulation of $PM_{2.5}$ concentration, the concentrations of OC and EC increased, and the proportion of $PM_{2.5}$ decreased (Ji et al., 2019). In a series of research reports on the evaluation of the Air Pollution Prevention and Control Action Plan (2013-2017), in the Chengdu-Chongqing region, the concentration of $PM_{2.5}$ gradually decreased from 2013 to 2015, and the proportion of NSA in $PM_{2.5}$ gradually increased, which also shows that when pollution is aggravated, the chemical composition of higher $PM_{2.5}$ concentrations is more complex, and the unknown component will contribute to a certain quality (Wang et al., 2019). We have revised the corresponding part in the manuscript. Now, it reads as follows:

This phenomenon occurs because some chemical components are included in the statistical analysis. It also reflects that the chemical components of $PM_{2.5}$ have more complex characteristics when pollution is aggravated. Some studies have analysed the changes in the chemical composition of particulate matter in regions with severe pollution in China in recent years, and the results show that the concentration of particulate matter has been significantly reduced, but other components (except NSA and carbonaceous aerosol) have higher contribution characteristics at higher particle concentrations (Geng et al., 2019;Wang et al., 2019). The variation trend of OC, EC and metal elements with increasing $PM_{2.5}$ concentration is similar to that of NSA (Fig. 2c), and this variation trend of OC and EC is consistent with the results of long-term observation research carried out in Beijing (Ji et al., 2019).

*Air Pollution 294 Prevention and Control Action Plan" Such discussion should be put in Sect 3.1.*

**Response:**

We appreciate your comments and apologize for our unprofessional description. We have adjusted the position of this discussion and revised the manuscript.
* * *
*Page 15 line 325 "As shown in Fig. S4, from 9:00 to 11:00 a.m., the concentrations of SO2, NOx, NH3, CO and other gases increased significantly, indicating that the primary emission of pollutants was relatively strong. At this time, higher RH (Fig S5) also provides favourable conditions for the formation of secondary aerosols and promotes the accumulation of NSA" But the RH in Fig. S5 is decreasing in contrast to an increase in aerosol concentrations?*

**Response:**

We appreciate your comments and apologize for our unprofessional description. As you have noticed, RH is indeed decreasing from 9:00 to 11:00. In our analysis, despite the decrease, RH was still relatively high (approximately 65%). Therefore, we have revised and supplemented this description. Now, it reads as follows:

As shown in Fig. S7, from 9:00 to 11:00 a.m., the concentrations of $SO_2$, NOx, $NH_3$ and CO increased significantly, indicating that the primary emission of pollutants was relatively strong. At this time, although RH is in a declining stage, it still has a relatively high atmospheric humidity (approximately 65%), and $O_3$ and $NO_2/NO$ also occasionally show an increasing trend, indicating that the atmospheric oxidizability has also increased (Figs. S7 and S8). This situation also provides favourable conditions for the formation of secondary aerosols and promotes the accumulation of NSA.
* * *
*Sect. 3.4.1 In general, it is true that the emissions of multi-pollutant may come from the same kinds of sources. But you cannot draw such a conclusion based on correlation studies. Because the variation of most pollutants, especially those of long-lifetime, is strongly influenced by the boundary layer developments, and may show a similar diurnal variation in spite of different origins (sources).*

**Response:**

We appreciate your comments and apologize for our unprofessional description. We agree with you. During the ACP discussion stage of the manuscript, our research team also carefully reviewed this text and found that the current analysis is not appropriate and may affect the integrity of the manuscript. Therefore, we decided to delete this section.
* * *
*Page 20 Line 430 "Figure 7 shows the variation characteristics of NSA chemical conversions and meteorological conditions with increasing RH. SOR and NOR increased with increasing RH, suggesting that $SO_2$ and $NO_2$ were more likely to produce sulfate and nitrate under higher RH conditions. In Fig. 7, how did you do the calculation, classifying the data according to RH or you keep all input the same but change RH only? In the former case, the apparent correlation with RH may not represent the real causation as chemical compositions and other parameters may change also change.*

**Response:**

We appreciate your comments and apologize for our unprofessional description. We have revised this text. In response to your concerns, we classified and statistically analysed the variation characteristics of NOR and SOR under different RH conditions according to RH observation data. Regarding the latter comment you raised, we also very much agree with you. The correlation is not enough to explain the relationship between chemical components and other parameters. Therefore, we combined the phase state of the chemical components analysed by the ISORROPIA-II thermodynamic equilibrium model to supplement the analysis.
* * *
*Sect 3.5.2 I understand that the authors adopted this approach based on a published study. This approach, however, is subject to several problems, e.g., neglecting the dilution of pollution in the course of transport which may overestimate the contribution of distant sources, or the endpoint is not necessary at the ground level, or why 24 hour (aerosols have a longer lifetime) etc. If you still want to keep this part, please explicitly*

*include these caveats in the text to avoid misinterpretation of this result.*

**Response:**

We appreciate your comments and apologize for our unprofessional description. As you pointed out, aerosols do influence of emissions, diffusion, chemical conversions and deposition in the process of regional transport. In our study, PSCF is a kind of conditional probability function relationship. Using Meteorological data from the National Oceanic and Atmospheric Administration (NOAA) to analyse the potential sources of pollution is helpful for explaining the importance of regional joint prevention and control measures for air pollution control. We choose a 24-hour simulation time, mainly considering the following factors. The aerosol spatial distribution characteristics show that there is a high concentration of pollutants in the Sichuan Basin in Southwest China (Gui et al., 2019), and due to the unique topography of the Sichuan Basin, air pollution is obviously affected by the internal emission (Qiao et al., 2019). We agree that the endpoints of the backward trajectory are not on the ground, in fact, as you think. The results of PSCF reflect the potential source of pollution in a plane, rather than the three-dimensional spatial structure characteristics, and the endpoints of the backward trajectory are reflected in the design plane grid, which also better reflects the high-value regional distribution features of PSCF. 
[revised manuscript text omitted]

decreasedhas a downward trend (Fig. 2a and b). This phenomenon occurs because some chemical components are included in the statistical analysis. It also reflects that the chemical components of $PM_{2.5}$ have more complex characteristics when pollution is aggravated. Some studies have analysed the changes in the chemical composition of particulate matter in regions with severe pollution in China in recent years, and the results show that the concentration of particulate matter has been significantly reduced, but other components (except NSA and carbonaceous aerosol) have higher contribution characteristics at higher particle concentrations(Geng et al., 2019;Wang et al., 2019). The variation trend of OC, EC and metal elements with increasing $PM_{2.5}$ concentration is similar to that of NSA (Fig. 2c), and this variation trend of OC and EC is consistent with the results of long-term observation research carried out in Beijing (Ji et al., 2019). When the $PM_{2.5}$ was less than 50 μg/m$^3$ and greater than 250 μg/m$^3$, the mass concentrations of NSA were 11.57 and 90.06 μg/m$^3$, respectively, and the proportions were 37.78 and 31.45% respectively. Comparing Fig. 2b and d, it was found that NSA was always the main contributor in the entire process of $PM_{2.5}$ accumulation, which was significantly higher than the proportions of OC and EC (Ji et al., 2019;Li et al., 2019b). In the accumulation process of $PM_{2.5}$ concentrations greater than 50 μg/m$^3$, $NO_3^-$ nitrate accounts for a high proportion in SNA and was is stable at approximately 14%, and the proportion of $SO_4^{2-}$ and $NH_4^+$ sulfate and ammonium continues to decrease (Li et al., 2019b;Wang et al., 2016). When the $PM_{2.5}$ concentration was less than 50 μg/m$^3$, the concentration of $SO_4^{2-}$ was higher than that of $NO_3^-$, and the concentration of $NH_4^+$ was lower than the $NH_4^+$ concentration of $PM_{2.5}$ at 50 to 100 μg/m$^3$, possibly due to $SO_4^{2-}$ sulfate concentration was higher than $NO_3^-$ nitrate, forming more chemically stable ammonium sulfate$(NH_4)_2SO_4$ (Guo et al., 2017a). In addition, when $PM_{2.5}$ was less than 50 μg/m$^3$, low RH and strong solar radiation were also important ways to generate sulfate (Yao et al., 2018).

**3.2 Monthly and seasonal variations**

The monthly variation characteristics of NSA from 2015 to 2017 are shown in Fig. 3. At the beginning and the end of each year, the pollutant concentration is relatively high and relatively low in the middle of each year. The meteorological conditions also have

obvious monthly variation characteristics (Fig. S5 a and b), from April to August, they have higher  WS and lower RH , which is not only conducive to the dilution and diffusion of pollutants but also reduces the chemical conversions of pollutants by aqueous phase and influencing the formation of secondary inorganic aerosols  (Wang et al., 2016;Ji et al., 2019). Overall, the concentrations are higher in January and December and lower in July and August. The highest monthly average  $NO_3^-$ reached 19.98 μg/m$^3$ in January 2017, and the highest monthly average of $SO_4^{2-}$ and $NH_4^+$ were 22.08 μg/m$^3$ and 12.66 μg/m$^3$ in January 2015, respectively. The lowest concentrations of NSA appeared in August 2017, which were 1.96, 3.07 and 1.62 μg/m$^3$. The gaseous precursors of NSA also have obvious monthly variations, and the NOx and $SO_2$ trends were similar to those of $NO_3^-$  and $SO_4^{2-}$  (Fig. 3 and S4). NH$_3$ emissions were significantly different, with increases in warmer months (April-July) and colder months (September-December). On the one hand, NH$_3$ volatilization was promoted by relatively high Ts (Fig. S5c); on the other hand, the use of agricultural fertilizers and livestock farming were also important sources of NH$_3$ in China. Second, from urban region, fossil fuel combustion and motor vehicle emissions also contribute significantly (Liu et al., 2013b;Pan et al., 2016). Notably, NH$_3$ increased significantly from April to December 2017 compared with 2015 and 2016, especially during low- T months (Fig. S4c). The result of an analysis of the monthly concentration variation of pollutants indicate that the implementation of pollution reduction and control measures should be strengthened at the beginning of each year (January to March) and the end of the year (October to December).

The seasonal variation in NSA was shown in Fig. S6, and the concentration in winter was much higher than that in summer. $NO_3^-$  only declined in spring and summer

from 2015 to 2017, with an increase in autumn and winter (Fig. S6a). Seasonal variations in  $NH_4^+$ were similar to those of , with higher concentrations in winter and the lowest in summer (Fig. S6c). This may be because higher  Ts and WSs not only can promote the decomposition of $NH_4NO_3$ in summer but also promote the dilution and diffusion of pollutant concentrations (Guo et al., 2017a;An et al., 2019). There is a significant downward trend in $SO_4^{2-}$, which continues to decrease in spring, summer and winter from 2015 to 2017 (Fig. S6b). In autumn, the concentration was the highest in 2016, and it was significantly lower in 2017 than in 2015 and 2016. The variation amplitude of NSA and gaseous pollutants in cold months was significantly higher than that in warm months (Figs. 3,  4 and S6). This higher variation amplitude may be due to the differences in pollutant accumulation and scavenging processes. This finding also indicates that the instability of local pollutant emissions and regional transport during cold months was affected by meteorological conditions (Li et al., 2017;Ji et al., 2018). The large variation amplitude of pollutants in different months, similar to the changes in the Beijing-Tianjin-Hebei region of northern China and Chengdu, are due to the accumulation and removal of pollution by meteorological conditions and pollutants emissions (Ji et al., 2019;Qin et al., 2019;Zhang et al., 2019a).


[Figure]

[Figure]

Fig. 3. Monthly variations in NSA (nitrate, sulfate and ammonium) concentrations from 2015 to 2017. (a) NO₃⁻. (b) SO₄²⁻. (c) NH₄⁺.

[Figure]

Fig. 4. Monthly variations in $NO_x$, $SO_2$ and $NH_3$ concentrations from 2015 to 2017. (a) $NO_x$. (
[revised manuscript text omitted]

[Figure]

**3.4. 1 Chemical conversion characteristics of SNA**

 Fig. 6 shows the abilities of  and $SO_2$, $NO_2$ to chemically convert to $NO_3^-$ and $SO_4^{2-}$  nitrates ts at different $PM_{2.5}$ concentrations. With the increase in $PM_{2.5}$ concentration,  NOR and  SOR gradually increased, indicating that the formation ability of  $NO_3^-$ and $SO_4^{2-}$  increased during the formation of air pollution. In this study, when the $PM_{2.5}$ concentration is $\leqslant$ 50 μg/m$^3$, the average of NOR and SOR are 0.07 and 0.27, respectively, and when the $PM_{2.5}$ concentration is greater than 250 μg/m$^3$, the average of NOR and SOR increase to 0.22 and 0.41, , respectively, indicating that the chemical conversion and formation ability of secondary inorganic aerosol was obviously enhanced when the air pollution was aggravated.

~~decreasing trend, as shown in Fig. 6a. With the accumulation of PM₂.₅ concentration, metal elements (Fe and Mn) also showed an increasing trend and were similar to SOR and NOR. Previous studies have shown that mineral dust elements such as Fe and Mn can play a catalytic role in the formation of atmospheric sulfate . The Pearson's correlation statistics of SOR and NOR with Fe and Mn under different PM₂.₅ concentration conditions are shown in Table S2; it is only under high PM₂.₅ concentration conditions (>200 μg/m³) that SOR and NOR have a positive correlation. This result is similar to those of previous studies in Beijing and Xi'an, where Fe and Mn play a limited catalytic role in sulfate formation . . SomeSORis11,there may be a photochemical reaction pathway leading to the conversion of SO₂ to sulfateFig. 6a shows that in addition to the photochemistry contributing to SO₂ oxidation, there may be a more important pathway leading to the conversion of SO₂ to sulfate.~~

[Figure]

Fig. 6. Analysis of atmospheric chemical conversion ability at different PM₂.₅ concentrations. (a)  NOR (nitrogen oxidation ratio). (b) SOR (sulfur oxidation

ratio)

 Fig. 7 shows the variation characteristics of NSA chemical conversions  with increasing RH.  NOR and  SOR increased with increasing RH, suggesting that  and  were more likely to produce NO$_3^-$ and SO$_4^{2-}$  under higher RH conditions. Previous studies have shown that in the presence of NH$_3$, NO$_2$ can promote the chemical conversion of SO$_2$ to SO$_4^{2-}$  in the aqueous phase (Wang et al., 2016). In an aerosol aqueous phase environment , alkaline aerosol (NH$_3$) components can promote the dissolution of SO$_2$ and formation of SO$_4^{2-}$  under the oxidation of NO$_2$ (Cheng et al., 2016). Especially when the atmosphere was polluted, the formation of SO$_4^{2-}$  by SO$_2$ through the aqueous phase environment can contribute most of the SO$_4^{2-}$  (Sun et al., 2013). When the RH is greater than 80%, the NOR appears to decline, possibly because HNO$_3$ is semivolatile, and the T increases at this time (Fig. 7 c), which is not conducive to the condensation of gaseous HNO$_3$ to the particulate matter, which affects the amount of NO$_3^-$ in PM$_{2.5}$ (Guo et al., 2017a). According to the ISORROPIA-II thermodynamic equilibrium model simulation, AWC  also increase with RH (Fig. 7 d), and the increase of AWC can provide a liquid environment for aerosol, 
[revised manuscript text omitted]

---

## Author Response (AR2)

**Response to Reviewer Comments**
* * *
**Title: Elucidating the pollution characteristics of nitrate, sulfate and ammonium in PM$_{2.5}$ in Chengdu, southwest China, based on three-year measurements**
* * *
**Minor revisions:**

*This concerns text on lines 262-275:*

*The sentence starting on line 265 "The phenomenon...". This is a confusing statement. You mean that this result occurs because not all chemical species are well constrained? This needs a bit more elaboration, so the reader can understand this.*

*This other point is: It seems that you refer to a study reporting lower organic fraction at high PM2.5, but in that case, one of the reviewer believes that the inorganic fraction is increased. It thus cannot explain the decrease of the inorganic fraction here. Could you please explain the reasons for the reduced inorganic fraction and what other chemical compounds (e.g., dust particle) may contribute to that.*

**Response:**

We appreciate your comments and apologize for our unprofessional description. We re-examined this description and found that the description was indeed inappropriate and prone to misunderstanding. With the accumulation of PM$_{2.5}$ concentration, NSA, OC, EC and metal element concentrations have an increasing trend, but their ratio with PM$_{2.5}$ gradually decreases, indicating that other components have a higher contribution. Studies have shown that the contribution of unknown components in PM$_{2.5}$ has a higher proportion, such as in Xi'an (29.5-38.2%) and Chengdu (16.5-33.8%), and under higher PM$_{2.5}$ concentration conditions, the proportion increases (Huang et al., 2014;Li et al., 2017). Also, under the implementation of the Air Pollution Prevention and Control Action Plan (2013-2017), the PM$_{2.5}$ concentration was significantly reduced in 2017 compared with 2013, and the contribution of unknown components of its chemical composition to PM$_{2.5}$ was reduced, such as eastern China (from 22% down to 20%), Beijing-Tianjin-Hebei (from 24% down to 23%) and the Yangtze River Delta (from 24% down to 22%) also show that the contribution of unknown components at higher PM$_{2.5}$ concentrations will increase (Geng et al., 2019). Therefore, our analysis results, on the one hand, can be attributed to those chemical compositions such as ions and mineral dust which are not included in the statistics(Huang et al., 2014;Zhang et al., 2015), and on the other hand, can be attributed to the contribution of unknown components.

So we revised the description in the manuscript, Now, it reads as follows:

[revised manuscript text omitted]
 PM2.5 decreased (Fig. 2a and b). This phenomenon occurs because some chemical components are included in the statistical analysis. It also reflects that the chemical components of PM2.5 have more complex characteristics when pollution is aggravated. Some studies have analysed the changes in the chemical composition of particulate matter in regions with severe pollution in China in recent years, and the results show that the concentration of particulate matter has been significantly reduced, but other components (except NSA and carbonaceous aerosol) have higher contribution characteristics at higher particle concentrations (
[revised manuscript text omitted]

---

## Author Response (AR3)

**Response to Editor**

Dear Editor,

We are very grateful for your recognition and acceptance.

We have a little content modification in this version, here to remind you.

In data quality assurance and quality control (QA/QC), the $NO_3^-$ and $NH_4^+$ data in Fig. S2 are redrawn, and the $SO_2$ data in Fig. S15 are redrawn. In Fig. 5, the text label (Spr, Sum, Aut and Win) has also been modified. The title of Fig. S10 is also modified, changing the "Box diagram" to "Box-plot", corresponding changes are also made on line 405 of the text. The data format of Table 2 has been adjusted and marked.

Besides, we checked and supplemented the DOI information of the references in the references.

We regret and regret the mistakes caused by our carelessness, and we guarantee that these changes will not affect the conclusions of this research.

We appreciate your help and time.

Sincerely yours,

Xingang Liu and coauthors.

School of Environment

Beijing Normal University

100875 Beijing China

E-mail: liuxingang@bnu.edu.cn; lxgstar@126.com

Tel: +86-13810193569